# Quantification of human contribution to soil moisture-based terrestrial aridity

Yaoping Wang [1,2,10], Jiafu Mao [2,10] ✉, Forrest M. Hoffman[3],
Céline J. W. Bonfils [4], Hervé Douville [5], Mingzhou Jin [1,6],
Peter E. Thornton [2], Daniel M. Ricciuto [2], Xiaoying Shi[2], Haishan Chen [7],
Stan D. Wullschleger[2], Shilong Piao [8] & Yongjiu Dai [9]

Current knowledge of the spatiotemporal patterns of changes in soil moisture-based terrestrial aridity has considerable uncertainty. Using Standardized Soil Moisture Index (SSI) calculated from multi-source merged data sets, we find widespread drying in the global midlatitudes, and wetting in the northern subtropics and in spring between 45°N–65°N, during 1971–2016. Formal detection and attribution analysis shows that human forcings, especially greenhouse gases, contribute significantly to the changes in 0–10 cm SSI during August–November, and 0–100 cm during September–April. We further develop and apply an emergent constraint method on the future SSI's signal-to-noise (S/N) ratios and trends under the Shared Socioeconomic Pathway 5-8.5. The results show continued significant presence of human forcings and more rapid drying in 0–10 cm than 0–100 cm. Our findings highlight the predominant human contributions to spatiotemporally heterogenous terrestrial aridification, providing a basis for drought and flood risk management.

Historical drying trends have been demonstrated to occur over the land surface, mostly in the subtropics and midlatitudes, through diverse indicators such as aridity and drought indices, the extent of drylands, and the frequency and severity of drought events[1–11]. Such drying trends are also widely projected to continue during the twenty-first century, especially under the high greenhouse gas emission pathways[1–6,9]. However, the detailed magnitudes, statistical significance, and spatial patterns of the drying trends depend considerably on the choice of specific drought or aridity indices[9,12,13], and on the sources of historical data (e.g., site measurements, satellite products, model outputs, or observational proxy data)[14–16]. Meteorological drought or aridity indices

based on precipitation and potential evapotranspiration are widely used[2,4,6,8,17], but their long-term trends are disconnected from the trends in vegetation growth or individual hydrological variables (e.g., leaf area index [LAI] and runoff)[9,12,18]. Also, the vast majority of past studies focused on Earth system model (ESM) simulations[1,3,5,6,10], or used one or a few data sets (e.g., satellite observations, reanalysis, offline model simulations)[2,7–9,11], resulting in high potential biases or uncertainty in drying or wetting except in a few hot-spot regions (e.g., Southern Europe, southern/southwestern North America, southern Africa)[15,19].

The causes of terrestrial drying can be understood in terms of the effects of natural internal variability (e.g., teleconnections), natural

[1]Institute for a Secure and Sustainable Environment, University of Tennessee, Knoxville, TN, USA. [2]Environmental Sciences Division and Climate Change Science Institute, Oak Ridge National Laboratory, Oak Ridge, TN, USA. [3]Computational Sciences and Engineering Division and Climate Change Science Institute, Oak Ridge National Laboratory, Oak Ridge, TN, USA. [4]Program for Climate Model Diagnosis and Intercomparison, Atmospheric, Earth, & Energy Division, Lawrence Livermore National Laboratory, Livermore, CA, USA. [5]Centre National de Recherches Météorologiques, CNRM/GMGEC/AMACS, Université de Toulouse, Météo-France, CNRS, Toulouse Cedex 01, France. [6]Department of Industrial and Systems Engineering, University of Tennessee, Knoxville, TN, USA. [7]Key Laboratory of Meteorological Disaster, Ministry of Education/Joint International Research Laboratory of Climate and Environment Change/Collaborative Innovation Center on Forecast and Evaluation of Meteorological Disasters, Nanjing University of Information Science & Technology, Nanjing, China. [8]Sino-French Institute for Earth System Science, College of Urban and Environmental Sciences, Peking University, Beijing, China. [9]School of Atmospheric Sciences, Sun Yat-sen University, Guangzhou, China. [10]These authors contributed equally: Yaoping Wang, Jiafu Mao. ✉e-mail: maoj@ornl.gov

solar variability and volcanic eruptions, and human-induced greenhouse gases and aerosol emissions. Previous studies[16,17,20] have effectively separated these effects by using formal detection and attribution (D&A) methods[21] on ESM predictions that have different combinations of these natural and anthropogenic factors turned on and off. However, these studies focused on either meteorological drought[17,20] or agricultural drought over part of the globe[16], and they did not reveal seasonal variations or vertical differences across soil layers. The D&A of seasonal multilayer soil moisture changes will add value to existing understanding of terrestrial drying because of the direct relevance of soil moisture to terrestrial biophysical and biogeochemical processes[9,22–24] and land-atmosphere feedbacks[25].

To date, the analysis of trends and the D&A analysis of soil moisture have been hampered by the limited availability of continuous, long-term, broadscale observations (e.g., multilayer soil moisture, soil moisture before the satellite era) and the low signal-to-noise (S/N) ratio of most water cycle changes[11,26–29]. We recently developed a set of long-term soil moisture data sets that were merged from a comprehensive list of observations, reanalysis, and offline model simulations and showed better performance than the source data sets[30]. These merged data sets can effectively reduce the potential biases and uncertainty caused by the limited sampling of source data sets in past studies[2,7–9,11]. In this study, we converted the merged data sets to a drought index (the 3-month Standardized Soil Moisture Index [SSI]; "Methods" and Supplementary Methods Sect. 1) and conducted a formal pattern-based D&A analysis[16,17,21] using, on one hand, the average value of the merged SSI as pseudo-observation and, on the other hand, the SSI of the latest Coupled Model Intercomparison Project Phase 6 (CMIP6) historical and single-forcing experiments[31,32]. We further developed a generalized additive model (GAM)-based emergent constraint method[11,33] to constrain the future S/N ratios of the D&A analysis[16,17,21] and the trends in the SSI under a high-emission scenario (Shared Socioeconomic Pathway [SSP] 5-8.5). In the D&A analysis, we followed the practice of previous D&A studies on hydrological variables[17,20,34–36] to enhance the S/N ratio by aggregating the pseudo-observed and simulated SSI to zonal averages. The zonal averaging reduces the influences from natural internal variability, which increases in importance at smaller spatial scales[37], and the less-understood local-scale forcings such as land use and land cover change[38], but retains the influences from large-scale circulations (e.g., the expansion of the Hadley cell[39,40], the shifting of the Intertropical Convergence Zone[17]). We conducted the analysis for each month and both the surface (0–10 cm) and root-zone (0–100 cm) layers to reveal the seasonal and vertical patterns of possible anthropogenic influences on the SSI changes.

## Results

### SSI trends in the pseudo-observation and CMIP6 simulations

The global average 3-month SSI time series showed consistent drying signals from 1971 to 2016 in the pseudo-observation and the CMIP6 ALL simulations (forced by all anthropogenic and natural forcing agents) during April–September and October–March, for the surface and root-zone soil layers (Fig. 1a, b, e, f). The differences in SSI between the average values of the brightening decades (1987–2016) and the end of the dimming decades (1971–1986)[41] ($\Delta$) ranged between $-3.57 \times 10^{-2}$ and $-1.54 \times 10^{-2}$, with larger $\Delta$ occurring in the surface soil layer than in the root-zone layer during April–September in both the pseudo-observation and the ALL simulations (Fig. 1a, b, e, f). In the ALL simulations, the global historical drying trends continued into the future and were greater in the surface soil layer than in the root zone (Fig. 1a, b, e, f).

Zonal patterns showed that the global average drying were mainly driven by the Northern Hemisphere between 20°N and 40°N and the Southern Hemisphere, where the pseudo-observation and the ALL simulations showed consistent drying trends in both the surface and the root-zone layers (Fig. 1c, d, g, h). The pseudo-observation

showed more wetting and less drying than the ALL simulations above 40°N, but these differences were mostly within the uncertainty caused by natural internal variability and structural differences across the CMIP6 ESMs (i.e., within the 95% confidence interval [CI] of the ALL simulations; Fig. 1c, d, g, h). The major differences (i.e., pseudo-observation outside the 95% CI of the ALL simulations) around 60°N in the Northern Hemisphere spring were caused by differences in eastern Canada and western Europe (Supplementary Fig. 2), which were likely related to the overestimation of the increasing trends in air temperature[42] and potential evapotranspiration in the ALL simulations (Supplementary Fig. 3). The pseudo-observation also had major differences from the ALL simulations in a few months near 30°N, and in February–July in 0°N–20°N (Fig. 1c, d, g, h). The former difference may be caused by the underestimated rate of the expansion of the Hadley cell[39,40] by the ALL simulations or model biases in the Monsoon-related precipitation in northern India (Supplementary Fig. 3; Singh et al.[43]). The latter may be caused by biases in the merged soil moisture data sets for the pseudo-observation in the Sahel region. The merged soil moisture data sets partially depended on the CERA20C, ERA20C, ERA-Interim, and ERA5 reanalyses[30], whose precipitation drivers all display negative biases in the temporal trends over the Sahel region compared to gridded rain gauge observations[44] (Supplementary Fig. 4).

### D&A analysis of the zonally averaged 3-month SSI

In this study, we used a pattern-based D&A method to investigate the influence of external forcings on the pseudo-observed SSI changes. In the first step, for each soil layer and each month of the year, we calculated the ALL fingerprint, which represents the mode-based spatial signature of SSI changes in response to the combination of anthropogenic and natural forcings ("Methods" and Supplementary Methods Sect. 2). The ALL fingerprint was defined as the leading empirical orthogonal function (EOF) of the multi-model average zonal-mean SSI anomalies over the 1971–2100 period, derived from the concatenated CMIP6 historical and future climate simulations forced by changes in all the anthropogenic and natural forcings (Supplementary Table 2). The combined month-latitude fingerprint patterns (Fig. 2a, e) resemble, for both soil layers, the month-latitude patterns of the ALL trends (Fig. 1d, h), indicating that the fingerprints captured the main spatiotemporal characteristics of the zonally averaged 3-month SSI changes. The greatest drying responses in the ALL fingerprints occurred in the summer season (i.e., June–August in the Northern Hemisphere and December–February in the Southern Hemisphere) (Fig. 2a, e), which were slightly shifted compared with the months with greatest drying in the ALL trends (Fig. 1d, h). To verify the seasonal shifts were not caused by difference in time periods employed to calculate the fingerprints, we calculated the fingerprints over 1971–2020 (ALL-2 fingerprints; Fig. 2b, f) instead of 1971–2100 and found similar seasonal shifts. In comparison with the ALL fingerprints (derived from 84 concatenated simulations), the GHG fingerprints (derived from 30 historical simulations forced by anthropogenic greenhouse gases only) showed more widespread drying trends, and the AER fingerprints (derived from 28 historical simulations forced by anthropogenic aerosols only) showed more wetting trends (Fig. 2c, g, d, h; Supplementary Table 2). These differences suggest that both the GHG and AER forcings influenced the ALL fingerprints. We note that during January–June, the AER fingerprints were quite different from the AER trends and displayed discontinuity compared to July–December (Fig. 2d, h; Supplementary Fig. 5). These were caused by the lack of clear directional changes in the AER-forced changes in SSI, as indicated by the mostly insignificant trends in the principal components associated with the AER fingerprints in these months (Supplementary Fig. 6).

The second step of the D&A method determines whether the ALL fingerprints were statistically detectable in the pseudo-observed SSI changes ("Methods" and Supplementary Methods Sect. 2). To achieve

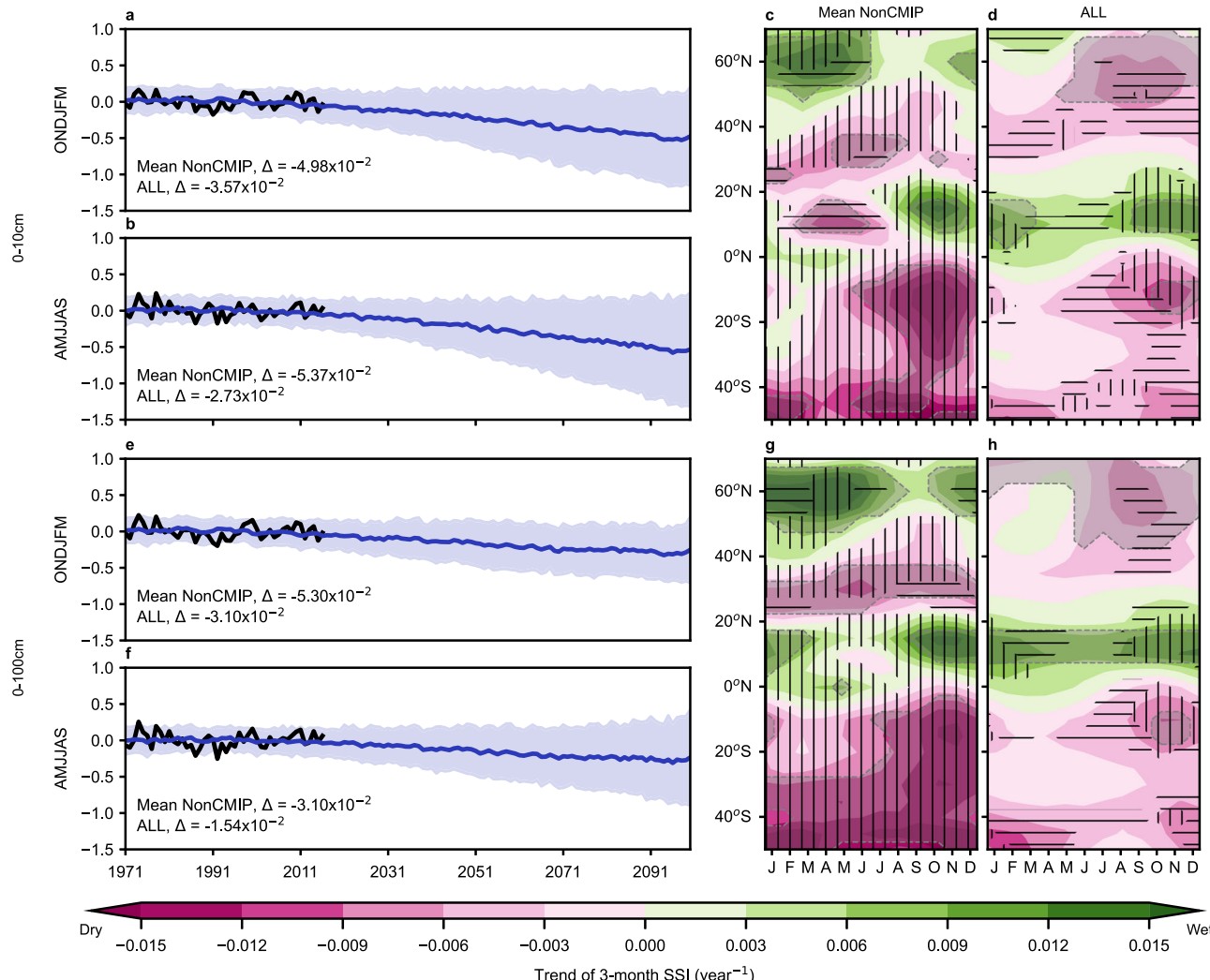

**Fig. 1 | Historical and future evolution of the 3-month Standardized Soil Moisture Index (SSI) in the 0–10 cm and 0–100 cm soil layers. a**, **b**, **e**, **f** The global mean time series of the October–March (ONDJFM) and April–September (AMJJAS) average SSI. Black lines represent the pseudo-observation (Mean NonCMIP). Blue lines and blue shading represent the average and 95% confidence intervals of the ALL simulations (forced by all anthropogenic and natural forcing agents). Δ represents the difference between the mean SSI of 1987–2016 and 1971–1986. **c**, **g** The month-by-latitude SSI trends over 1971–2016 of the pseudo-observation. Vertical hatching indicates where the trends had the same signs as the average trends of the ALL simulations. Horizontal hatching indicates where the trends had the opposite signs to the average and were outside the 95% confidence intervals of the trends of the ALL simulations. The month abbreviations are, from left to right (J, F, M, A, J, J, A, S, O, N, D), in the order of January–December. Gray shading indicates where the trends were significantly different from zero at 95% confidence level. **d**, **h** The month-by-latitude SSI trends over 1971–2016 of the ALL simulations, averaged over all the models. Vertical hatching indicates at least 90% of the ALL simulations agreed on the signs of the trends, and horizontal hatching 80%. Gray shading indicates where more than 50% of the ALL trends were significantly different from zero at 95% confidence level. The month abbreviations are, from left to right (J, F, M, A, J, J, A, S, O, N, D), in the order of January–December.

this goal, for each soil layer and each month, we projected the pseudo-observed zonal SSI during 1971 to 2016 onto the ALL fingerprint, resulting in a 46-year time-series that reflected the spatial agreement between the ALL fingerprint and the pseudo-observed SSI through time. The trend in this time series constituted the 46-year pseudo-observed signal. We also projected the zonal SSI of the concatenated control runs simulations, which were only influenced by natural internal variability, onto the ALL fingerprints. Note that the SSI patterns from the control simulations should not resemble the ALL fingerprint, except by chance. Using all the overlapping chunks of 46-year segments in the projected control time series, we calculated a probability distribution of unforced 46-year trends. We considered a pseudo-observed signal to be detectable at a 95% confidence level when the signal lay outside the two-sided 95% CI of the unforced trends, which indicates that the signal is very unlikely to result from internal variability alone. In the last step of the D&A, for each soil layer and each month, we projected the SSI from historical simulations

under 6 different sets of forcing agents onto the ALL fingerprint. We then calculated the 1971 to 2016 trends and obtained the 6 probability distributions of simulated forced signals. We considered a detected signal to be attributable to a specific set of forcing agents if the pseudo-observed signal lay within the 95% CI of the distribution of the correspondingly forced simulated signals. The 6 sets of forcing agents we considered were: ALL, GHG, AER, ANT (anthropogenic forcings only), GHGAER (anthropogenic greenhouse gases and aerosols only), and NAT (natural solar and volcanic forcings only) (Supplementary Table 2).

We describe here the results of this investigation. For the surface soil layer, the 1971–2016 pseudo-observed signals were detectable from August to November. During those months, the pseudo-observed signals were within the 95% CIs of the distributions of the ALL, ANT, GHGAER, and GHG-forced signals and thus attributable to those sets of forcings (Fig. 3h–k). Those detected pseudo-observed signals were also often in better agreement with (i.e., closer to the mean of) the ALL

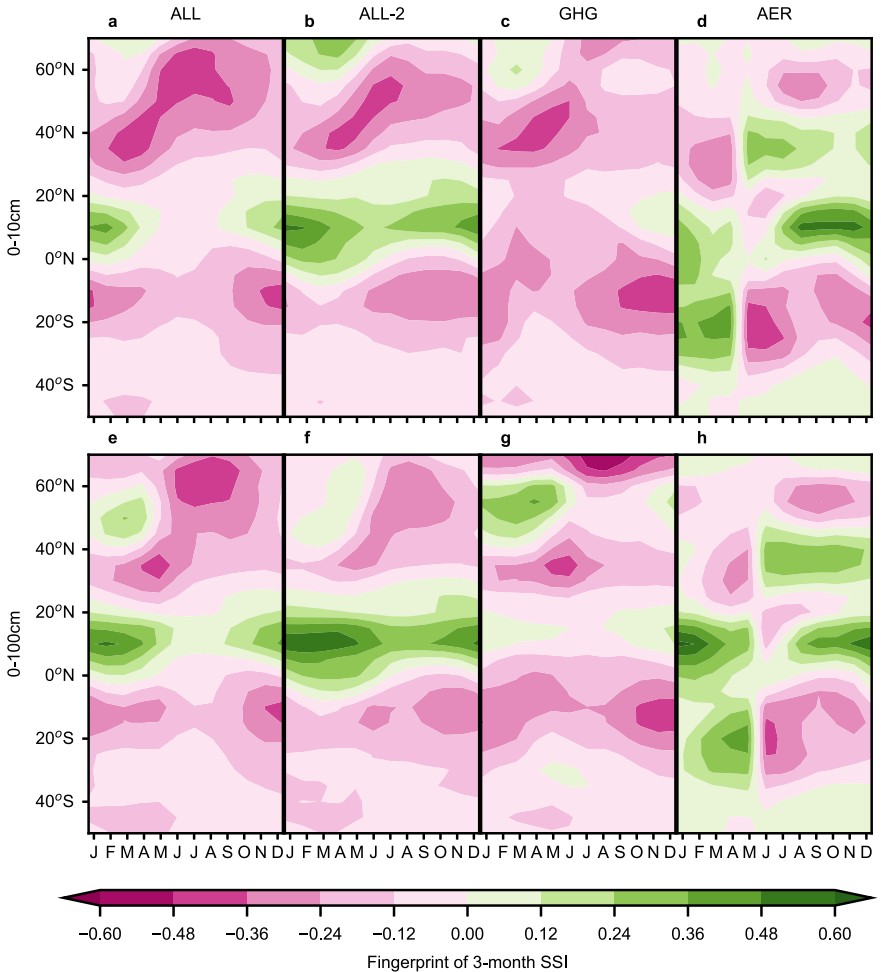

**Fig. 2 | Month-by-latitude fingerprints of the CMIP6 simulations under different forcings for the 3-month Standardized Soil Moisture Index (SSI) in the 0–10 cm and 0–100 cm soil layers. a–h** The forcings abbreviations are: ALL and ALL-2—forced by all anthropogenic and natural forcing agents, GHG—forced by anthropogenic greenhouse gases only, AER—forced by anthropogenic aerosols only. The ALL fingerprints were calculated on zonally averaged 3-month SSI over the 1971–2100 period, and the ALL-2, GHG, AER fingerprints over the 1971–2020 period. The month abbreviations are, from left to right (J, F, M, A, J, J, A, S, O, N, D), in the order of January–December.

and ANT distributions than with the GHGAER and GHG distributions (Fig. 3i–k). The surface pseudo-observed signals could not be attributed to the NAT or AER forcings (Fig. 3a–k). For the root-zone soil layer, the pseudo-observed signals were detectable and attributable to the ALL, ANT, GHGAER, and GHG forcings in September–April (Fig. 3m–p, u–x). The pseudo-observed signals could not be attributed to the NAT forcings during those months, nor to the AER forcings in October–December and March–April (Fig. 3m–p, u–x). In September, January, and February, the pseudo-observed signal was located within the 95% CI of the distribution of the AER-forced signals, but its position near the upper tail of the distribution did not warrant formal attribution. In a nutshell, the soil moisture signals were detectable and attributable to the ALL forcing (but also to the ANT, GHGAER, and GHG forcings) for a 4-month period (from August to November) for the surface layer, and a longer 8-month period (September to April) for the root-zone layer.

To quantify the relative strengths of the detected pseudo-observed signals in each month, we summarized the temporal evolution of the pseudo-observed signals using the concept of detection time[45] ("Methods"). The earliest detection time occurred in October for the surface layer and in December for the root-zone layer (Table 1). The latest detection time occurred in August for the surface layer and in January and March for the root-zone layer (Table 1). These results suggested that the detected anthropogenic influences had the

strongest presence in the pseudo-observation in autumn, and weaker presence in summer and spring.

Using the ALL fingerprint, we further conducted systematic sensitivity analysis on the D&A by altering the timescale, distributional fit, and soil moisture data sets used in calculating the SSI, and the set of model ensemble members for calculating the fingerprints, signals, and unforced trends (Supplementary Methods Sect. 3). Although the months in which the pseudo-observed signals were detected, and the detection times of those signals varied slightly, the results generally support our conclusions (Fig. 3, Table 1) (Supplementary Tables 4–7).

## Evolution of the anthropogenic signals in the twenty-first century

Following the D&A on historical signals, we investigated the future presence of anthropogenic forcings using the future S/N ratios of the ALL simulations under the Shared Socioeconomic Pathway 5-8.5 (SSP5-8.5). In those S/N ratios, the signals (S) refer to the simulated ALL-forced signals, obtained as trends in 46-year moving windows in the ALL simulations during 1971–2100. The noise (N) refers to the standard deviation of all the 46-year unforced trends obtained from the projected control run time series. We considered the future S/N ratios to mean significant presence of anthropogenic forcings when the absolute values of those S/N ratios were 1.96 or greater, which was consistent with the detection criterion for the pseudo-observed signal, if

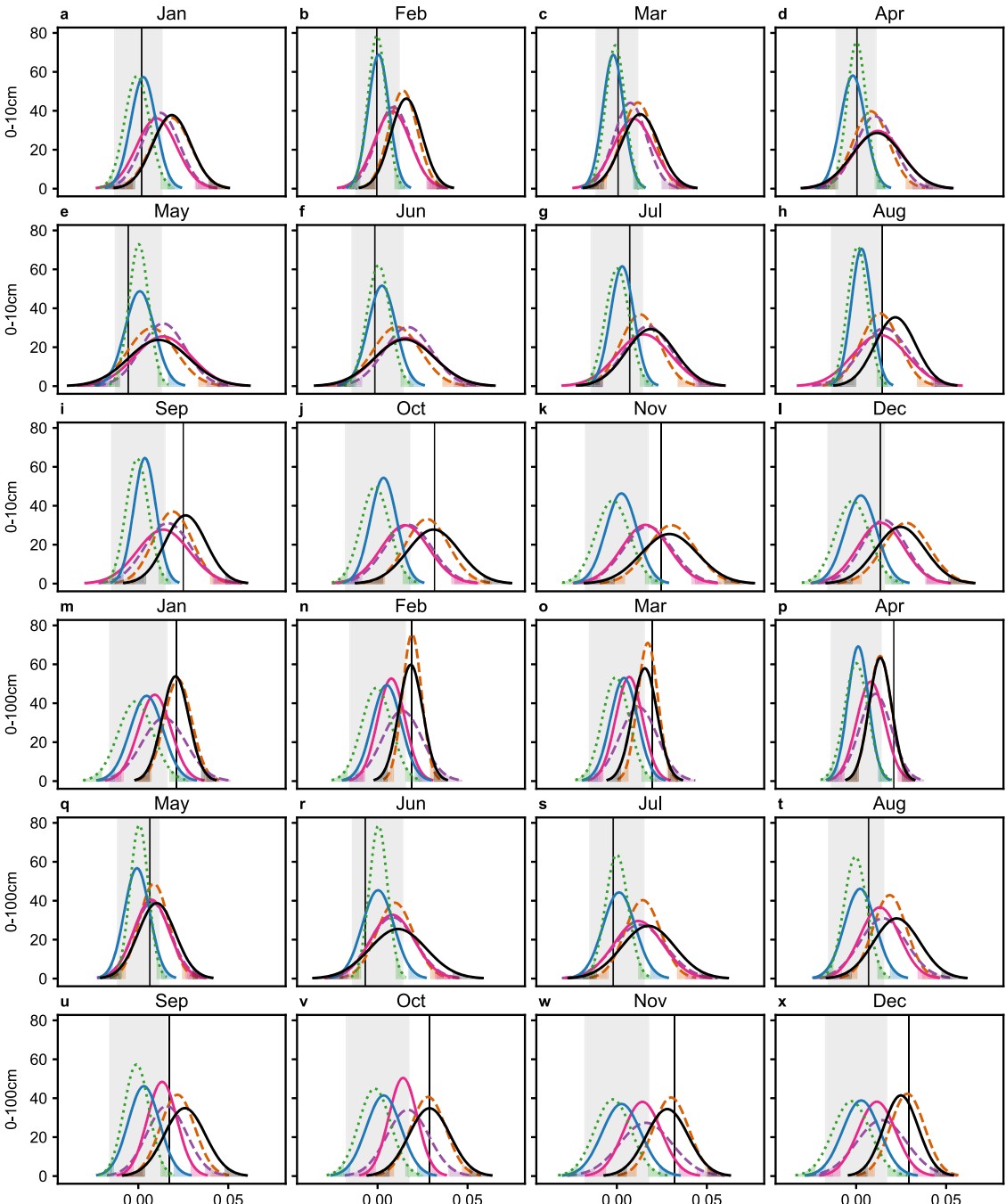

**Fig. 3 | Monthly detection and attribution of the pseudo-observed signals over 1971–2016 in the 0–10 cm and 0–100 cm soil layers. a–x** Vertical black lines represent the pseudo-observed signals. Gray shaded vertical regions represent the 95% confidence intervals of the unforced trends. Bell-shaped lines represent the fitted Gaussian distributions on the simulated signals of the various forced CMIP6 simulations: ALL (black solid, forced by all anthropogenic and natural forcing agents), ANT (brown dashed, forced by anthropogenic forcings only), GHGAER (purple dashed, forced by anthropogenic greenhouse gases and aerosols only), GHG (magenta solid, forced by anthropogenic greenhouse gases only), AER (blue solid, forced by anthropogenic aerosols only), NAT (green dotted, forced by natural solar and volcanic forcings only). Shading beneath the bell-shaped lines represent the two-sided 95% confidence intervals of the distributions. All the signals and unforced trends were on the ALL fingerprints and were for the same window length (46 years).

**Table 1 | Detection times at which the pseudo-observed signals on the ALL fingerprints became significant at the 95% confidence level**

| Month | Jan | Feb | Mar | Apr | May | Jun | Jul | Aug | Sept | Oct | Nov | Dec |
|---|---|---|---|---|---|---|---|---|---|---|---|---|
| 0–10 cm | — | — | — | — | — | — | — | 2015 | 2002 | 1995 | 2005 | — |
| 0–100 cm | 2013 | 2014 | 2013 | 2009 | — | — | — | — | 2012 | 1997 | 2000 | 1994 |

Em dashes (—) indicate no significant signals.

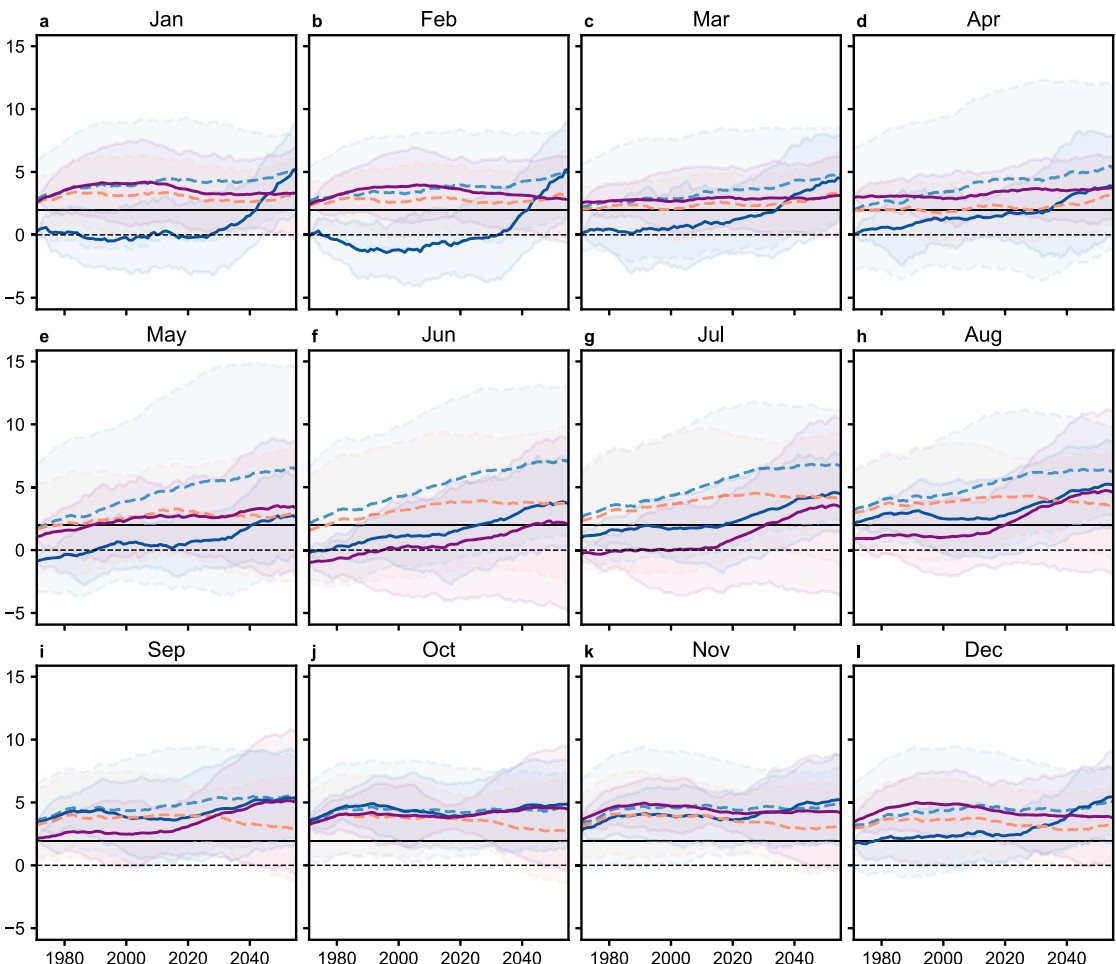

**Fig. 4 | Evolution of the raw and constrained signal-to-noise (S/N) ratios over the future periods 1971–2016, 1972–2017, …, 2055–2100. a–l** The x-axis indicates the starting years of the future periods. Dark blue (purple) solid lines represent the constrained S/N ratios of 0–10 cm (0–100 cm). Shaded area bound by dark blue (purple) solid lines represent the two-sided 95% confidence intervals of the constrained S/N ratios of 0–10 cm (0–100 cm), which were calculated using regression results, not the Gaussian assumption. Light blue (orange) dashed lines represent the average of the raw S/N ratios of 0–10 cm (0–100 cm). Shaded area bound by light blue (orange) dashed lines represent the two-sided 95% confidence intervals of the raw S/N ratios of 0–10 cm (0–100 cm). Horizontal solid lines indicate the 1.96 threshold, above which the S/N ratios mean significant presence of anthropogenic forcings at 95% confidence level. The signals in the S/N ratios were ALL-forced (i.e., forced by all anthropogenic and natural forcing agents), on the ALL fingerprints, and for 46-year window lengths. The noises in the S/N ratios were defined as the standard deviation of the unforced trends on the ALL fingerprints for the 46-year window length.

one assumes that the unforced trends followed Gaussian distribution (Supplementary Methods Sect. 2.5–2.6).

The raw future ALL-forced S/N ratios were likely too high, considering that the average historical ALL-forced signals were often too high compared with the pseudo-observed signals (Fig. 3), and that the ESMs-based estimated noises and the ALL-forced signals have a common source of error, i.e., the same model physics. To avoid drawing too strong conclusions about the future presence of anthropogenic forcings, we constrained the raw future S/N ratios using a GAM-based emergent constraint method ("Methods"). Emergent constraint is a well-accepted approach to reduce the uncertainty in future climate projections by bringing in information from historical observations while making use of the modeled historical-future relationships[11,46,47]. Here, the historical observations were the pseudo-observed S/N ratios, and the modeled historical-future relationships were between the historical and future S/N ratios of the ALL simulations. The detailed formula and physical justifications of the specific implementation of emergent constraint here are in the "Methods" and Supplementary Methods Sect. 4.

The constrained simulated S/N ratios of the surface soil layer did not become significant at the 95% confidence level (i.e., did not exceed ±1.96) in December–July until the 2000s or later, but were generally significant in August–November (Fig. 4). The constrained S/N ratios of the root-zone layer were mostly significant in September–May but were not significant in June–August until 2020 or later (Fig. 4). These seasonality patterns were consistent with the historical seasonality in the detection times of the pseudo-observed signals (Table 1). The constrained S/N ratios of the surface soil layer increased over the future period in all the months, suggesting increased aridity, whereas the constrained S/N ratio of the root-zone soil layer only showed increases in May–August, and showed fluctuations in the other months (Fig. 4).

We further analyzed the implications of constraining the future S/N ratios on the zonal trends in SSI using mathematical relationships between the S/N ratios and the trends (Supplementary Methods Sect. 5). For the surface soil layer, the adjusted future SSI trends based on the constrained S/N ratios had smaller drying magnitudes, especially during January–June, than the un-adjusted average SSI trends of the ESMs (compare the last row to the first row of Supplementary Fig. 10). However, even after emergent constraint, the surface 3-month SSI showed accelerating drying trends over the course of the twenty-first century for nearly all the latitudes and months of the year (last row of Supplementary Fig. 10). For the root-zone SSI, the adjusted future SSI trends based on the constrained S/N ratios had smaller drying

trends than the un-adjusted mean SSI trends in general and had reversal from drying to wetting trends in June–July in the Southern Hemisphere (compare the last row to the first row of Supplementary Fig. 11). The accelerating wetting around 50°N during January–March and accelerating drying during October–April in the northern high latitudes and southern mid-latitudes in the root-zone SSI were consistent between the adjusted and un-adjusted future SSI trends (Supplementary Fig. 11).

## Discussion

Using multi-source merged soil moisture data sets[48] and the CMIP6 simulations[31,32], we verified the previously reported historical and future drying of the global land surface[1–8,13–15,49,50] and further demonstrated clear vertical, zonal, and seasonal patterns. The changes in the SSI in the two soil layers were less widespread than previously reported changes in potential evapotranspiration[51], which is consistent with past findings about the overestimation of drought by potential evapotranspiration changes[12,13]. The surface SSI dried more rapidly than the root-zone SSI, as can be seen in the historical differences between the brightening and dimming decades during April–September (the Δ values in Fig. 1a, b, e, f), the future time series under the SSP5-8.5 scenario (Fig. 1a, b, e, f), the future evolutions of S/N ratios (Fig. 4), and the future zonal trends especially around 50°N in the Northern Hemisphere spring (Supplementary Figs. 10–11). One potential cause of these vertical divergences is that the surface soil responds faster than the root-zone to meteorological conditions because of the concentration of plant roots in the surface soil and the slow speed of capillary rise[27,52]. In the Northern Hemisphere spring, the impacts of the increasing atmospheric evaporative demand can be exacerbated on the surface SSI by the decreasing snow cover but mitigated on the root-zone SSI by the relatively low spring vegetation activity. This contrasting effect of snow cover was supported by the fact that the correlations between snow water equivalent and SSI had opposite signs between the surface and the root-zone soil layers in the Northern Hemisphere spring (Supplementary Fig. 12). Also, reduced stomatal conductance in response to drought and the future increase in atmospheric carbon dioxide concentration[53] mitigates the impact of rising temperatures on transpiration, which affects both the surface and root-zone soil moisture; but no such mitigation effect exists for soil evaporation, which mainly affects the surface layer. This mechanism is supported by the higher correlations between air temperature and the surface SSI than the root-zone SSI (Supplementary Fig. 12). Under severe drought, vegetation die-off may also exacerbate the drying of the surface SSI due to less shading.

Although the historical and future SSI trends and the fingerprints of the ALL simulations indicated mostly drying of the two soil layers, the underlying mechanisms varied by months and latitudes. In the northern mid- to high latitudes (20°N and above), the seasonal drying pattern was mainly driven by strong increases in air temperature, and secondarily by increasing leaf area index in summer and early autumn and decreasing snow water equivalent in spring (Supplementary Fig. 13). In the northern subtropics (0°N–20°N), increases in precipitation was the main contributor to the summer wetting (Supplementary Fig. 13). In the Southern Hemisphere, increases in temperature was the main contributor to the drying, followed by increasing leaf area index and decreasing snow water equivalent below 40°S (Supplementary Fig. 13).

The detection of significant differences of the pseudo-observed signals from natural internal variability and the attribution of the detected signals to greenhouse gases-dominated anthropogenic forcings (ALL, ANT, GHGAER, and GHG) in the surface 3-month SSI in August–November and in the root-zone 3-month SSI in September–April (Fig. 3) expanded the conclusions of previous D&A research on drought[11,16,17,20,28,29]. Whereas the previous studies demonstrated significant anthropogenic impacts on the annual or summer

(either Northern Hemisphere summer or locally defined) average drying, we demonstrated seasonally varying impacts that were the greatest in the transition months (August–November) but also occurred in the root-zone in the boreal winter and spring months (Fig. 3). These seasonal patterns suggest that the traditional focus on annual or summer average drying may lead to underestimation of the drought risks in late summer and autumn, which can still affect ecosystem functions and reservoir operations[54,55], and neglection of the increase in flood risks in the northern high-latitude spring (Fig. 2). Some previous D&A studies showed that the AER forcing impacted annual and summer drought before the 1980s[16,17]. This study could not reliably attribute the detected signals to the AER forcing because the pseudo-observed signals were very near the upper edge of the 95% CI of the AER-forced signals (Fig. 3). This finding may be because much of the studied historical period (1971–2016) was post-1980. Considering the regionally non-uniform changes in aerosol emissions[56], the complexity of aerosol effects, and current model inadequacies[57], future studies are needed to better detect, attribute, and understand the mechanisms of AER-forced soil moisture changes.

In recognition of the biases in the S/N ratios of the ALL simulations compared to the pseudo-observation (Fig. 3), we developed a GAM-based emergent constraint approach to better estimate the simulated future S/N ratios than the average of the ESMs ("Methods"). We further quantified the implications of constraining the future S/N ratios on the future SSI trends using mathematical relationships between the S/N ratios and the SSI trends (Supplementary Methods Sect. 5). Bias-correcting the future projection based on historical D&A results is a common practice in optimal fingerprinting-based D&A[58,59], but the practice cannot be applied to pattern-based D&A[16,17,21] because of different mathematical formulas. The developed GAM-based emergent constraint approach is similar to conventional emergent constraint in that it estimates linear relationships between modeled historical and future variables[11,46,47]. But the traditional linear regression framework of emergent constraint[11,46,47] requires a separate linear regression to be fitted per future period, while a single GAM can be fitted on all the future periods and ensures that the constrained future S/N ratios form a smoothly varying time series ("Methods"). A limitation of GAM is that it can only test the overall significance of the fitted term[60] and cannot test whether the linear relationship between historical and future S/N ratios becomes less significant over time (e.g., reported by Winkler et al.[47] because of nonlinear physical relationships). Therefore, future studies should explore more advanced statistical testing methods to find potential temporal changes in the significance of the emergent relationship. Despite this limitation, the GAM reconciled the difference between the modeled and the pseudo-observed S/N ratios, was parsimonious, and achieved nearly always better fit on the future S/N ratios than fitting a separate linear regression per future period (Supplementary Fig. 8). Therefore, the developed approach provides a reasonable framework for pattern-based D&A studies to account for the model biases in future S/N ratios.

Apart from the GAM framework, additional limitations exist in the current study. We only used the leading fingerprint in the D&A and the emergent constraint, but some anthropogenic signals may exist in other fingerprints[17]. The pseudo-observation contains some bias in the Sahel region (Supplementary Fig. 4) because of the limitations of the source data sets. The CMIP6 ESMs do not fully or consistently simulate interactive atmospheric chemistry[31], dynamic vegetation[61], and land use and land cover change[62]. These limitations should be addressed with future methodological and data advancements.

In summary, we identified significant human contributions to global SSI-based drying of the surface soil in August–November and the root-zone soil in September–April over 1971–2016. The drying mainly occurred in the northern and southern mid-latitudes, and in the summer and autumn seasons in the northern high-latitudes; counteracting wetting occurred in the northern subtropics and in spring in the

northern high-latitudes. The anthropogenic impacts were mainly contributed by greenhouse gas emissions. Pseudo-observation constrained future S/N ratios and SSI trends under the SSP5-8.5 scenario suggested accelerating drying in the surface soil and in the root-zone soil, except in the spring in the northern high latitudes, where the root-zone SSI showed accelerating wetting. These heterogeneous SSI changes point to greater risks of drought and floods in the future, suggesting the need for latitude- and seasonally dependent mitigation and adaptation measures. By revealing detailed spatiotemporal patterns of human forcings' impacts on long-term SSI changes, this study advanced current understanding of the changes and causes of terrestrial aridity, and generated results and methodological developments that will be of interest to the scientific community and the broader public.

## Methods

We used the SSI for the D&A rather than the raw soil moisture because the magnitudes of soil moisture in ESMs are highly dependent on model-specific assumptions about soil properties, evapotranspiration, runoff, and drainage, whereas the temporal variabilities are more robust[63]. For the D&A analysis in the main text, we calculated the surface and root-zone SSI at the timescale of 3 months using a distributional fit procedure that involved the Gaussian mixture distribution (Supplementary Methods Sect. 1). The calculated 3-month SSI of each month reflects the average soil moisture conditions of the current month and previous two months (e.g., the 3-month SSI in February reflected the average soil moisture between previous year's December and this year's January and February). The pseudo-observed SSI, abbreviated as Mean NonCMIP and spanning 1971–2016, was the average SSI derived from three merged soil moisture products[30] that are independent of the CMIP5 or CMIP6 ESMs (Supplementary Methods Sect. 1). We calculated the modeled SSI under the influences of all the anthropogenic and natural forcings (ALL), anthropogenic greenhouse gases only (GHG), anthropogenic aerosols only (AER), combined influences of anthropogenic greenhouse gases and aerosols (GHGAER), anthropogenic forcings only (ANT), natural solar and volcanic forcings only (NAT), and internal natural variability only (piControl) using all the available appropriate CMIP6 ensemble members (Supplementary Table 2). For sensitivity analysis on the D&A results, we also calculated monthly SSI at the 1- and 6-month timescales, using alternative pseudo-observations, statistical distribution, and CMIP6 ensemble members (Supplementary Methods Sect. 3). We aggregated all the pseudo-observation and modeled SSI to 5° zonal averages for the D&A.

We conducted the D&A analysis separately for the SSI in each month of the year and soil layer used a pattern-based method[16,21,35,45,64] (Supplementary Methods Sect. 2). We calculated the ALL fingerprint as the first empirical orthogonal function (EOF) of the multi-model average of zonal-mean SSI anomalies derived from the concatenated CMIP6 historical and SSP5-8.5 simulations. We then projected the zonal-mean SSI from the pseudo-observation and from the CMIP6 simulations forced by different sets of forcing agents (ALL, GHG, AER, GHGAER, ANT, NAT) onto the ALL fingerprint. We treated the 1971–2016 trend in the projected pseudo-observed time series as the pseudo-observed signal (referred to as S). We treated the 1971–2016 trends in the projected time series under a specific set of forcing agents (ALL, GHG, AER, GHGAER, ANT, or NAT) as the simulated forced signals. The probability distribution of those simulated forced signals was fitted on the model ensemble members forced by the same set of agents (Supplementary Table 2). We also projected the piControl simulations onto the ALL fingerprint and calculated the unforced trends of all the overlapping 46-year segments in the projected time series. We calculated the noise (referred to as N) as the standard deviation of those unforced trends. When a pseudo-observed signal was outside the two-tailed 95% confidence interval

(CI) of the fitted probability distribution of the unforced trends, we considered the pseudo-observed signal detectable at the 95% confidence level. This corresponded to an absolute value of the pseudo-observed S/N ratio of 1.96 or greater, under the assumption of Gaussian distribution of the unforced trends. If, in addition, a detected pseudo-observed signal lay within the 95% CI of the distributions of the ALL, GHG, AER, GHGAER, ANT, or NAT-forced signals, we considered detected signal to be attributable to the indicated set of external forcing agents.

We used the concept of detection time[45] to quantify when the pseudo-observed signal first became detectable. For each month and soil layer, instead of calculating the pseudo-observed and ALL-forced signals as trends over a fixed time period (1971–2016), we calculated time-varying signals, with the starting year of the trends being in 1971, and the ending year varying between 1981 and 2016. For each time period, we also calculated the corresponding noise as the standard deviation of the unforced trends over an equal length of time. That is, the corresponding noise of the signals over 1971–1981 would be based on 11-year segments of the projected piControl series, over 1971–1982 based on 12-year segments, …, over 1971–2016 based on 46-year segments. The detection time was the ending year at which the time-varying pseudo-observed signal first became and afterward remained significant at the 95% confidence level[45]. To ensure that the detected pseudo-observed signals were attributable to the external forcings represented by the ALL fingerprint, we also set a consistency condition: if a detected signal was outside the 95% CI of the distribution of the ALL-forced signals over the same time period, then the signal was treated as if insignificant.

We constrained the biased future S/N ratios of the CMIP6 ALL simulations using a generalized additive model (GAM)-based emergent constraint method. The standard emergent constraint method estimates linear regression relationships between modeled historical and future variables, treating the historical-future pair of each individual ESM as an x–y pair in the regression[11,46,47,65]. If the slope of the linear regression is statistically significant, the observed historical variable is plugged into the regression equation to generate a constrained future value, which is presumably better than the average of the ESMs, and an uncertainty interval for the constrained future value[11,46,47,65]. In this study, the modeled historical variable was the historical ALL-forced S/N ratios over 1971–2016, the modeled future variable was the ALL-forced S/N ratios over various future time periods (1972–2017, 1973–2018, …, 2055–2100), and the historical observation was the pseudo-observed S/N ratio over 1971–2016. The linear regression approach only allows one future value to be generated with one regression. Therefore, if we used the linear regression approach, we would perform a separate linear regression between each pair of future and historical ALL-forced S/N ratios. Since the regression coefficients of any two adjacent future periods would be estimated separately, the constrained future S/N ratios of these two periods could differ considerably. Such discontinuity would violate the temporal smoothness of the S/N ratios as trends, for the trends over two substantially overlapping time periods (e.g., the 46-year time periods 1972–2017 and 1973–2018 have 45 overlapping years) should only differ slightly, unless the non-overlapping years were substantial outliers. To ensure smooth transition between the constrained S/N ratios of adjacent future periods, we used GAM[66], instead of linear regression, to estimate the historical-future relationships. The GAM takes the form $y = \beta_1 + s(x, t) + \varepsilon$, where $\beta_1$ is the intercept, $\varepsilon$ is the fitting residual, $y$ is the future modeled S/N ratio of an ESM, $x$ is the historical modeled S/N ratio of the same ESM, $t$ is the year, and $s(\bullet)$ is a tensor product smooth over $x$ and $t$—which, intuitively, is the sum of products between all the pairs of the marginal spline basis of $x$ and the marginal spline basis of $t$[60]. We set the marginal spline of $x$ to consist of one intercept term and one linear term of $x$. This setup ensured that the relationship between $x$ and $y$ was always linear, following the convention of past emergent constraint

studies[11,46,47,65]. We set the marginal spline of $t$ to have cubic order and determined the number of splines by minimizing the Akaike Information Criteria[67], because preliminary analysis showed that the linear regression coefficients between the historical and future S/N periods varied nonlinearly over time. We fitted a separate GAM for each month of the year using the PyGAM package[67]. We also averaged the ALL-forced S/N ratios of each ESM within its ensemble members before putting the values into the regression to prevent the ESMs with more ensemble members from exerting more influence on the regression. Supplementary Fig. 7 shows an example of such fitted tensor product $s(x, t)$. The GAM achieved nearly always better fit on the future ALL-forced S/N ratios than fitting a separate linear regression for each future period (Supplementary Fig. 8). All the fitted GAMs were significant at the 95% confidence level. In additional to statistical significance, the emergent constraint approach requires the modeled historical-future relationship to be physically justified[11,46,47,65]. Supplementary Methods Sect. 4 discusses the physical justification for this emergent constraint between the historical and future ALL-forced S/N ratios.

To propagate the effect of emergent constraint on the S/N ratios to ALL-forced zonal trends in SSI, we decomposed the zonal trends into the sum of two terms (Supplementary Methods Sect. 5). Briefly, the first term is proportional to the future S/N ratio, and the second term is interpreted as a remainder term that is dependent on the non-leading empirical orthogonal functions that were obtained as part of the D&A process. To adjust the SSI, we replaced the future S/N ratio in the first term with the constrained S/N ratio and kept the other terms in the equation the same.

Throughout this paper, all mentions of average values, the percentages of models that agreed in sign with the average value, and the percentages of models that were significant at the 95% confidence level should be understood as weighted. That is, the values were first averaged over the ensemble members of each ESM and then averaged over the ESMs, to prevent the ESMs with more ensemble members from dominating the results. Because agreement in sign and significance at 95% confidence level were Boolean values, "true" was treated as 1, and "false" as 0, in the weighted averaging. Standard deviations were not weighted and were calculated directly on the ensemble members of all the ESMs. Unless specified otherwise, all the 95% CI in this paper were calculated using the Gaussian assumption, i.e., equal to the weighted average $\pm 1.96 \times$ standard deviation. Also, unless specified otherwise, all the trends in this paper were calculated using linear least squares.

### Reporting summary
Further information on research design is available in the Nature Research Reporting Summary linked to this article.

## Data availability
The merged soil moisture data sets used to derive the pseudo-observed SSI in this paper are available at https://doi.org/10.6084/m9.figshare.13661312.v1. The soil moisture and soil moisture-drivers data of the CMIP6 ESMs under various external forcings were downloaded from https://esgf-node.llnl.gov/.

## Code availability
The source codes for this study are available at https://bitbucket.org/ywang11/soil_moisture_da/src/master/.

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

## Acknowledgements

This research was supported by an Oak Ridge National Laboratory subcontract no. 4000169153 (Y.W., M.J.), the Reducing Uncertainties in Biogeochemical Interactions through Synthesis and Computation Science Focus Area (RUBISCO SFA) funded through the Regional and Global Model Analysis activity in the Earth and Environmental Systems Sciences Division of the Biological and Environmental Research office in the DOE Office of Science (J.M., F.M.H., X.S.), the Lawrence Livermore National Laboratory contract no. DE-AC52–07NA27344 and the DOE Regional and Global Model Analysis Program under the Program for Climate Model Diagnosis & Intercomparison Science Focus Area (PCMDI SFA) (C.J.W.B.), the National Natural Science Foundation of China grant no. 42130609 (H.C.) and no. U1811464 (Y.D.). The authors thank Dr. Aurélien Ribes for his insightful suggestions on the initial version of this manuscript and Dr. Rongfan Chai for his aid in preparing some of the

graphs. This research used resources of the Compute and Data Environment for Science at ORNL. ORNL is managed by UT-Battelle, LLC, for DOE under Contract No. DE-AC05-00OR22725. This manuscript has been authored by UT-Battelle LLC under Contract No. DE-AC05-00OR22725 with the US Department of Energy (DOE). The US government retains and the publisher, by accepting the article for publication, acknowledges that the US government retains a nonexclusive, paid-up, irrevocable, worldwide license to publish or reproduce the published form of this manuscript, or allow others to do so, for US government purposes. DOE will provide public access to these results of federally sponsored research in accordance with the DOE Public Access Plan (http://energy.gov/downloads/doe-public-access-plan).

## Author contributions

J.M. conceived the research; Y.W. and J.M. performed the analyses and drafted the figures; Y.W. and J.M. wrote the first draft of the manuscript; F.M.H., C.J.W.B., H.D., M.J., P.E.T., D.M.R., X.S., H.C., S.D.W., S.P., and Y.D. reviewed and edited the manuscript before submission. All authors (Y.W., J.M., F.M.H., C.J.W.B., H.D., M.J., P.E.T., D.M.R., X.S., H.C., S.D.W., S.P., Y.D.) made substantial contributions to the discussion of content.

## Competing interests

The authors declare no competing interests.
