## [Peer review file · Nature Communications]

Reviewers' comments:

Reviewer #1 (Remarks to the Author):

Although the impacts of anthropogenic climate change on drought have been extensively studied, most were focused on meteorological drought indices (e.g., PDSI and aridity index), and few were on soil moisture, especially the long-term multi-layer soil moisture changes. The general methodology of this study appears to be robust, with the use of multiple observation-constrained soil moisture datasets, latest CMIP6 sensitivity simulations, and formal detection and attribution (D&A) analyses. Moreover, its demonstration of seasonally and vertically varying anthropogenic effects is very interesting and has implications for improving drought predictability and management. Considering the importance of soil moisture D&A to land surface processes and land-atmosphere feedbacks, the findings of the study are a significant contribution to both the D&A and drought research fields. I thus recommend accepting the manuscript after addressing the following comments and concerns.

1) SSI trends in the pseudo-observation and CMIP6 simulations. Observations showed that a drying tendency existed in the northern subtropics. However, the authors suggested that wetting occurred in the northern subtropics, which seems contrary to the results shown in Fig.1. Besides the anthropogenic warming effects that increase potential evaporation, this observed drying trend may be partially associated with the poleward extension of Hadley circulation. However, the multi-model means showed little drying tendency in the northern subtropics and the core drying region shifted further north. This makes me wonder whether current CMIP6 models have enough skill to reproduce the observed changes of the northern branch of the Hadley circulation.

2) Line 106: The authors stated that the temporal variability of model-based soil moisture tended to be more robust than its magnitude. However, they did not evaluate the models' performance in reproducing the observed spectrum structure of soil moisture temporal variability, such as the these at interdecadal, interannual and intraseasonal scales.

3) Detection and attribution. The authors used a complex D&A method to quantify the impacts of various external forcings on the spatiotemporal patterns of SSI changes. However, they did not add much explaining possible physical/biophysical mechanisms through which these external forcings pose influences on the surface and root-zone SSI. At least, the mechanisms behind GHG-induced SSI drying would be different from those of aerosols and volcanic activities.

4) Suppl. Lines 231-233: I do not think that is a good answer for the contradict signals of volcanic aerosol emissions in ALL and NAT PCs. The authors may need to explore further this 'significant' contradiction.

5) Fig. S5: It's unclear to me what has caused the large differences between the PCs associated with the main and jackknife-based fingerprints of ANT in Jan. and NAT in Jun.

Reviewer #2 (Remarks to the Author):

This paper conducts a detection and attribution analysis on merged soil moisture data products with an aim to quantify human contribution to soil moisture-based droughts. Here, the authors first converted soil moisture data products that are related to observations and that are simulated by CMIP6 models under various external forcing into standardized soil moisture index (SSI), and then compared them using pattern-based detection and attribution method. They claim that human influences affected soil moisture of both top 10cm and top 100cm soil layers in various 3-month seasons. The author developed, what they considered to be the first, observation-constrained future SSI projections. While it is interesting to compare observed soil moisture with those simulated by models to gain understanding of possible human influence on soil moisture, thereby attributing changes in drought (as the title of the paper may imply), there are a numbers of papers already on the same or similar subject. It is unclear to me if this paper brings sufficient novelty in methodology or our general understanding of soil moisture response to external forcing that would warrant it to publish in Nature Communications. Compared with existing literature, Bonfils et al. (2020) in particular, I am not sure we learnt a lot new here. In addition to this general observation, I also have a number of concerns regarding data, methods, presentation and interpretation.

1) The title: The title is misleading. The authors did the analysis on the standardized soil moisture index (SSI). While SSI is used in drought monitoring, only the low tail of SSI has something to do with drought. This paper focuses on trend of SSI, or long-term changes in mean of SSI; it does not deal with changes in extreme negative values of SSI (that indicate drought).

2) Data: The merged soil moisture product is perhaps the most comprehensive soil moisture data with regard to spatial and temporal coverage. But it is unclear if this data product is suitable for the analysis of long-term changes and thus if it fits for the purpose of this study in particular. The paper describing the development of this datasets is still under review according to ESSD website. I did read the discussion paper and realized that while the data developer argued add values of the data products in terms of "improvement and harmonized spatial, temporal, and vertical coverages", they did not conduct careful check of data homogeneity. Thus the suitability of the data product for the analysis of long-term change is not known. Given that the data are merged from different sources, one would assume inhomogeneity in the data product.

3) Methods: The pattern-based detection and attribution method is the same (but a bit simplified) as that used by Bonfils et al. (2020). But there are indications that the method may not have been implemented properly. For example, while Bonfils et al. (2020) detected human influence at the 5% level, this paper would state to have detected at the 95% confidence interval. Fig. S3 of this paper shows fingerprint for different forcings but the fingerprints for individual forcings have never been used in the detection as the signal is the projection of observations to the fingerprint of all forcing. The detection and attribution analysis is conducted on latitudinal average of soil moisture index without giving clear justification. Figure S10 seems to contradict the wisdom/justification of making latitudinal average: trends of ALL forcing simulations are of opposing signs over large continuous areas in the same latitude bands. The authors claimed they had invented a new emergent constraint based on S/N without giving a clear physical reasoning or proof, but only to walk back later that they didn't know if that emergent constraint actually worked (see discussion on page 18, in particular lines 438-440).

4) Presentation: The main text is relatively easy to understand, though there are so conceptual errors in describing results (e.g. detection at the 95% confidence interval etc.). But the Supplementary Information is impossible to follow. I read for more than 5 times and I am still not sure I fully understand how the authors did the analyses (I am sorry if I have misunderstood your work in my review).

Reviewer #3 (Remarks to the Author):

Dear Authors,

From my first reading of the manuscript, I feel that this manuscript is very interesting but currently not suited for publication in Nature Communications. The manuscript is well written and I see the importance but the current form of the manuscript needs more than major revisions in it's current form. The topic of D&A of soil moisture trends is very relevant and interesting, certainly with the CMIP6 models included. I feel the analysis is relevant but need further improvement. Please find some suggestions below.

I find the manuscript hard to read, it is difficult to follow the different steps, approaches, and experiments. The manuscript would benefit from a smaller set of experiments or a better description of the different experiments in a concise manner. The current manuscript balances on a lot of different analysis, different angles and it is very hard to detangle the main message. I find it very tough to get through the current form of the manuscript. I know and see that there is a lot of interesting results in

the manuscript, but the current content load of the manuscript is way too high. My recommendation would be to focus on 2/3 main message to better show the reader the added value.

Are these trends significant, please provide uncertainty bounds? If I read the caption of Figure 1, they are not outside of the 95% CI. Please also clarify left and right orientated hatching, better to use hatching and some horizontal lines. I left-orientated from left to right or right to left?

Line 147-148 is not very satisfying, if the authors indicate potential model deficiencies it would be better to show that this is the case instead of leaving the reader without any proof.

Line 151-153, another speculation that is not really support by any evidence. I would prefer

Mention how stomate closure response affect potential evaporation rates in the future

I feel that 22 extra figures in the appendix and 12 more table do not improve the readability of the manuscript. I feel that the authors should be able to provide the main message of the manuscript with a significant lower number of figures in the appendix.

Line 170, dot is at the wrong spot before the brackets

Reviewer #1 (Remarks to the Author):

Although the impacts of anthropogenic climate change on drought have been extensively studied, most were focused on meteorological drought indices (e.g., PDSI and aridity index), and few were on soil moisture, especially the long-term multi-layer soil moisture changes. The general methodology of this study appears to be robust, with the use of multiple observation-constrained soil moisture datasets, latest CMIP6 sensitivity simulations, and formal detection and attribution (D&A) analyses. Moreover, its demonstration of seasonally and vertically varying anthropogenic effects is very interesting and has implications for improving drought predictability and management. Considering the importance of soil moisture D&A to land surface processes and land-atmosphere feedbacks, the findings of the study are a significant contribution to both the D&A and drought research fields. I thus recommend accepting the manuscript after addressing the following comments and concerns.

1) SSI trends in the pseudo-observation and CMIP6 simulations. Observations showed that a drying tendency existed in the northern subtropics. However, the authors suggested that wetting occurred in the northern subtropics, which seems contrary to the results shown in Fig.1. Besides the anthropogenic warming effects that increase potential evaporation, this observed drying trend may be partially associated with the poleward extension of Hadley circulation. However, the multi-model means showed little drying tendency in the northern subtropics and the core drying region shifted further north. This makes me wonder whether current CMIP6 models have enough skill to reproduce the observed changes of the northern branch of the Hadley circulation.

Response: Thank the reviewer for the encouragement and comments. Regarding the northern subtropics issue, we performed a literature search and found that the current CMIP6 models underestimated the rate of expansion of the Hadley cell compared with reanalysis, although the CMIP6 models had a higher rate than the CMIP5 models (Hu et al., 2018; Xia et al., 2020). Such model biases may have caused underestimation of the drying tendency in 20°N–40°N, and the difference from pseudo-observation at this latitude. Based the global map of SSI (Fig. S1), we also suggest that the difference between the pseudo-observation and the ALL simulations was caused by the Indian Monsoon region. Therefore, we added the following explanation in the main text lines 133–138 (tracked change version lines 214–234): “The pseudo-observation also had major differences from the ALL simulations in a few months near 30°N, and in February–July in 0–20°N (Fig. 1c, d, g, h). The former difference may be caused by the under-estimated rate of the expansion of the Hadley cell by the ALL simulations^{39,40} or model biases in the Monsoon-related precipitation in northern India (Fig. S1; ref.⁴³).”

The overestimation of drying in the ALL simulations in 40°N–60°N was mainly contributed by the eastern Canada and western Europe regions (Fig. S1), and is likely related to the overestimation of the warming trends by the CMIP6 models in these regions (see Fig. 3 of (Fan et al., 2020)). Our additional analysis in the revised manuscript suggests that the potential evapotranspiration is overestimated in these regions in the ALL simulations (Fig. S2). Therefore, we added the following explanation in the main text lines 129–133 (tracked change version lines 210–214): “The major differences (i.e., pseudo-observation outside the 95% confidence interval [CI] of the ALL simulations) around 60°N in Northern Hemisphere spring were caused by differences in eastern Canada and western Europe (Fig. S1), which was likely related to the over-

estimation of the increasing trends in air temperature⁴² and potential evapotranspiration in the ALL simulations (Fig. S2).”

2) Line 106: The authors stated that the temporal variability of model-based soil moisture tended to be more robust than its magnitude. However, they did not evaluate the models’ performance in reproducing the observed spectrum structure of soil moisture temporal variability, such as the these at interdecadal, interannual and intraseasonal scales.

Response: Thank you for this suggestion. In the new version of our dataset paper (Wang et al., 2021), we performed a comparison between the power spectral densities of the merged and the source data sets using regional averages. Here, we reproduced the graph, but only showing the same ALL simulations of CMIP6 ESMs as used in the current D&A paper (Figure A). It can be clearly seen that the three products used to create the pseudo-observation in the main text, Mean ORS, OLC ORS, and EC ORS, had power spectral densities that were generally within the envelopes of the ALL simulations. This indicates that the selected CMIP6 soil moisture simulations did not systematically over- or underestimate the variabilities across different temporal scales. The auxiliary data sets, EC CMIP5, EC CMIP6, EC CMIP5+6, and EC ALL, which were used for sensitivity analysis in the Supplementary Material, also had power spectral densities that were mostly within the envelopes of the ALL simulations.

Figure A. Power spectral densities of the regional mean soil moisture of the merged data sets (Mean ORS, OLC ORS, EC ORS, EC CMIP5, EC CMIP6, EC CMIP5+6, EC ALL), and the ALL simulations (Source data sets). The power spectral densities were calculated on each individual ALL simulation, and the envelope shows the maximum and the minimum values of the calculated densities at each frequency. The regions are the IPCC SREX regions (Field et al., 2012). N – north, W – west, C – central, E – east, S – south.

3) Detection and attribution. The authors used a complex D&A method to quantify the impacts of various external forcings on the spatiotemporal patterns of SSI changes. However, they did not add much explaining possible physical/biophysical mechanisms through which these external

forcings pose influences on the surface and root-zone SSI. At least, the mechanisms behind GHG-induced SSI drying would be different from those of aerosols and volcanic activities.

Response: Thank you for raising this issue. We added mechanisms related discussion in the revised manuscript in lines 255–286 (tracked change version lines 485–654), and created some supporting figures (Figs. S5 and S9).

4) Suppl. Lines 231-233: I do not think that is a good answer for the contradict signals of volcanic aerosol emissions in ALL and NAT PCs. The authors may need to explore further this ‘significant’ contradiction.

Response: Thank you for pointing this out. We deleted the ANT, GHGAER, and NAT PCs from the paper because they were not used in the D&A in any way; also, the other two reviewers commented that the manuscript was cramped with information. Regarding the “significant” contradiction, we examined the original figures (reproduced in Figure B and Figure C below) more closely, and found that the contradiction mainly occurred in the months July to November. In these months, the NAT fingerprints were almost the opposite of the ALL fingerprints (Figure D). Therefore, the projection of the same volcanic aerosol-induced SSI patterns in these months on the NAT fingerprints would likely have opposite signs compared with the projection on the ALL fingerprints.

Figure B. The principal components associated with the ALL (1971–2100), ALL-2 (i.e., the ALL forcing but only using the simulations for 1971–2020), ANT, GHG, AER, GHGAER, and NAT forcing fingerprints.

Figure C. The principal components associated with the ALL (1971–2100), ALL-2 (i.e., the ALL forcing but only using the simulations for 1971–2020), ANT, GHG, AER, GHGAER, and NAT forcing fingerprints.

Figure D. The monthly fingerprints of the NAT and the ALL forcings.

5) Fig. S5: It's unclear to me what has caused the large differences between the PCs associated with the main and jackknife-based fingerprints of ANT in Jan. and NAT in Jun.

Response: Thank you for raising this issue. For the same reason as mentioned in comment #4, the ANT, GHGAER, and NAT fingerprints and PCs were removed from the new manuscript, and any uncertainty therein does not affect the D&A analysis. Therefore, we did not investigate into the differences.

Reviewer #2 (Remarks to the Author):

This paper conducts a detection and attribution analysis on merged soil moisture data products with an aim to quantify human contribution to soil moisture-based droughts. Here, the authors first converted soil moisture data products that are related to observations and that are simulated by CMIP6 models under various external forcing into standardized soil moisture index (SSI), and then compared them using pattern-based detection and attribution method. They claim that human influences affected soil moisture of both top 10cm and top 100cm soil layers in various 3-month seasons. The author developed, what they considered to be the first, observation-constrained future SSI projections. While it is interesting to compare observed soil moisture with those simulated by models to gain understanding of possible human influence on soil moisture, thereby attributing changes in drought (as the title of the paper may imply), there are a number of papers already on the same or similar subject. It is unclear to me if this paper brings sufficient novelty in methodology or our general understanding of soil moisture response to external forcing that would warrant it to publish in Nature Communications. Compared with existing

literature, Bonfils et al. (2020) in particular, I am not sure we learnt a lot new here. In addition to this general observation, I also have a number of concerns regarding data, methods, presentation and interpretation.

Response: Thank the reviewer for your comments. As cited in the Introduction of the main text, there are indeed many papers focused on the subject of drought, a few of which are already related to the D&A of drought. However, many of these past papers were mainly centered on the use of meteorological drought indices (Schlaepfer et al., 2017; Dai et al., 2018; Cook et al., 2014; Feng and Fu, 2013; Bonfils et al., 2020). As shown in Figure 1 of Yang et al. (2019), the meteorological drought indices derived from precipitation and potential evapotranspiration generally overestimated global drying tendencies; only the Palmer Drought Severity Index (PDSI), which has a more complicated formula of calculation, showed considerably similarity to surface soil moisture; but the PDSI still could not reveal detailed drought structures across soil layers. Compared with the meteorological droughts, soil moisture more directly affects the availability of water to plants. For example, past studies have demonstrated that soil moisture is a greater control on plant activity than vapor pressure deficit (Liu et al., 2020), and that multi-layer soil moisture can better explain vegetation growth than single layer (Li et al., 2020). Soil moisture also directly modulates the primary productivity, phenology, and land-atmosphere feedbacks (Fu et al., 2021; Lian et al., 2021; Gevaert et al., 2018).

Also, the majority of the past papers were either only focused on the analysis of ESM simulations (Xu et al., 2019; Lu et al., 2019; Cook et al., 2014; Greve and Seneviratne, 2015) or the use of limited observations, reanalysis, or offline model simulations (Schlaepfer et al., 2017; Deng et al., 2020; Feng and Fu, 2013; Lian et al., 2021; Douville and Plazzotta, 2017). Because soil moisture can be considerably different across data sets (Liu et al., 2019; Xu et al., 2021; Cheng et al., 2017), having different sources of products merged using strict statistical methods can overcome the biases/errors contained in individual data sets. Such data set-related improvement and novelty has been demonstrated in this manuscript and our related paper (Wang et al., 2021).

Additionally, the seasonal and layer-wise differences, identified by the present study but omitted in past studies, have significant practical implications. For example, because the drying was generally greater in summer than winter, the annual average drying examined by previous studies would exclude or underestimate the impacts of soil drying on seasonal vegetation dynamics (e.g., phenology). Also, the focus on annual averages by past studies could preclude the identification of spring wetting in the northern high-latitudes, which can significantly affect flood risks, reservoir operations, and vegetation productivity.

Therefore, although previous studies and this research address global aridification, our work adds more robustness and evidence, showing that human signals significantly existed in long-term changes of soil moisture-based drought and not just in meteorological drought indices (e.g., Bonfils et al., 2020); moreover, this research systematically clarifies evident seasonal and layer-wise variations in these human signals. We thus revised the entire Introduction (lines 55–107; tracked change version 58–178) to better address these advancements over previous studies.

Regarding Bonfils et al. (2020) specifically, they employed a meteorological drought index, Climate Moisture Index (CMI), which is based on precipitation and potential evapotranspiration. Therefore, it may likely share similar abovementioned deficiencies as to the meteorological drought indices. Moreover, Bonfils et al. (2020) applied their D&A methods on the joint changes in temperature, rainfall, and aridity; consequently, the signals and signal-to-noise ratios shown in their Figures 1, 2, and 3 reflected joint changes in temperature, rainfall, and aridity, containing some information from all three variables. Using only the SSI, our results are thus focused on the clean information directly from soil moisture, and have different physical meanings from those of Bonfils et al. (2020). For example, comparing their joint CMI fingerprints 1 and 2 (Extended Data Figs. 1c and 1f of Bonfils et al. [2020]) with the ALL fingerprint shown in our Figure 2, one can see that our fingerprint is different from any of their fingerprints, although a weighted combination of the two could probably yield more similar results to ours. Drying above 60°N only existed in fingerprint 1 of Bonfils et al. (2020) and was smaller than the drying in the southern hemisphere; but in our fingerprints, the >60°N drying is greater than that in the southern hemisphere. No wetting above 30°N existed in the fingerprints of Bonfils et al. (2020), but our fingerprints show wetting between 40°N and 60°N in spring. Between 0°N and 20°N, fingerprint 1 of Bonfils et al. (2020) showed drying, and fingerprint 2 showed wetting; but our fingerprints show that the wetting and drying depend on the season, especially in the 0–10 cm layer.

Because another reviewer also pointed out that our study lacked detailed discussion of physical mechanisms, we added new analysis and discussion about the influences of precipitation, temperature, LAI, and snow water equivalent on the SSI in the revised manuscript (see Discussion lines 255–286 [tracked change version lines 485–654], and Figs. S5 and S9). However, such additional analysis is unlikely to significantly improve the understanding of meteorological drought changes because LAI and snowmelt are not explicitly in their calculation formulas, although some influences may exist through land-atmosphere feedbacks. Thus, these analysis can be used to further distinguish our findings from Bonfils et al. (2020), which focused on the investigation of large-scale circulation mechanisms (e.g., shift in the ITCZ) underlying hydroclimate changes.

1) The title: The title is misleading. The authors did the analysis on the standardized soil moisture index (SSI). While SSI is used in drought monitoring, only the low tail of SSI has something to do with drought. This paper focuses on trend of SSI, or long-term changes in mean of SSI; it does not deal with changes in extreme negative values of SSI (that indicate drought).

Response: Thank you for point out the issue. We changed the title from “drought” to “terrestrial aridity”.

2) Data: The merged soil moisture product is perhaps the most comprehensive soil moisture data with regard to spatial and temporal coverage. But it is unclear if this data product is suitable for the analysis of long-term changes and thus if it fits for the purpose of this study in particular. The paper describing the development of this datasets is still under review according to ESSD website. I did read the discussion paper and realized that while the data developer argued add values of the data products in terms of “improvement and harmonized spatial, temporal, and vertical coverages”, they did not conduct careful check of data homogeneity. Thus the suitability

of the data product for the analysis of long-term change is not known. Given that the data are merged from different sources, one would assume inhomogeneity in the data product.

Response: Thank you for the comment. In the process of revising the data set paper (Wang et al., 2021), we performed statistical test of the temporal homogeneity of the merged data sets on a grid-by-grid basis using a previous procedure (Su et al., 2016). The new results are displayed in the Figure 4 and Figures S6 and S7 of the revised data set paper. The new analysis shows that in terms of mean soil moisture values, little inhomogeneity could be detected; in terms of the variance of the soil moisture values, inhomogeneity could be detected in 10%–30% of the grid cells, depending on which merged data set was evaluated against which reference data set. The highest percentage of inhomogeneity in variance was actually detected in a merged data set that was averaged from seven TRENDY v7 land surface models (see Table S1–S3 of the data set paper, column “Used time period,” which shows that only these seven models were used for the 1970–2016 period). This merged data set should not contain any inhomogeneity because offline land surface models and the simple averaging process do not contain inhomogeneity. We concluded that the merged data sets were homogeneous, and the detected homogeneity may have arisen from inhomogeneity and disturbance representation issues in the reference data sets as described in the data set paper (Wang et al., 2021). Therefore, we suggest that the merged data sets are suitable for D&A.

3) Methods: The pattern-based detection and attribution method is the same (but a bit simplified) as that used by Bonfils et al. (2020). But there are indications that the method may not have been implemented properly. For example, while Bonfils et al. (2020) detected human influence at the 5% level, this paper would state to have detected at the 95% confidence interval. Fig. S3 of this paper shows fingerprint for different forcings but the fingerprints for individual forcings have never been used in the detection as the signal is the projection of observations to the fingerprint of all forcing. The detection and attribution analysis is conducted on latitudinal average of soil moisture index without giving clear justification. Figure S10 seems to contradict the wisdom/justification of making latitudinal average: trends of ALL forcing simulations are of opposing signs over large continuous areas in the same latitude bands. The authors claimed they had invented a new emergent constraint based on S/N without giving a clear physical reasoning or proof, but only to walk back later that they didn’t know if that emergent constraint actually worked (see discussion on page 18, in particular lines 438-440).

Response: Thank you for the pointing out these issues. During the process of creating this manuscript, we had a lot of conversations with one of our coauthors, Dr. Bonfils, who was also leading Bonfils et al. (2020), ensuring that the procedure was correctly implemented. The 5% significance level and 95% confidence level/interval are different terminologies that refer to the same thing—when a test statistic is within the 95% CI, there is no statistical significance; when outside, there is statistical significance at a 5% level (total probability is always 100%, hence $100\% - 95\% = 5\%$). We changed the terminologies to $p = 0.05$ and $p = 0.01$ to remove the confusion (main text lines 167–168, 177–178, 194, 197, 220, 223, 591, 599, 618–619, 628, 631 [tracked change version lines 300–302, 312–313, 347–349, 444–447, 1130, 1138, 1157–1158, 1169, 1172]; Supplementary Material lines 177). In the original Fig. S3, some of the fingerprints were indeed not used in the detection. We deleted the entire figure to avoid confusing the readers in

the updated version because the necessary fingerprints (ALL, GHG, AER) are already presented in the main text.

Latitudinal averaging is a common practice in D&A on hydrological variables enhancing the S/N ratio, such as precipitation (Sarojini et al., 2016; Marvel and Bonfils, 2013; Zhang et al., 2007), evapotranspiration (Douville et al., 2013), and joint changes in precipitation, temperature, and drought index (Bonfils et al., 2020). Latitudinal averaging has also been performed in other analyses on global temperature and soil moisture to highlight large-scale model performance or pattern changes (e.g., Santer et al., 2018; Berg et al., 2017). All of these variables have longitudinal heterogeneity. In general, the relative importance of natural internal variability to external forcings becomes larger at smaller scales (Bindoff et al., 2013). Also, external forcings other than greenhouse gases and aerosols, such as land use and land cover change, can be more important at smaller scales (Lawrence et al., 2016); however, the simulated system responses to these forcings remain quite uncertain because of inconsistent or incomplete representations among different ESMs. Latitudinal averaging would remove the influences from local factors and longitudinal factors, such as Monsoon circulations, but not from latitudinal factors, such as the expansion of the Hadley cell, the shifting of ITCZ, and the pole-to-equator gradient in warming. Therefore, we considered the latitudinal averaging not to invalidate the D&A process.

In recognition of the existence of longitudinal heterogeneity, we performed an analysis that shows historical S/N ratios at the 5° grid level (Figure E, Figure F). The S/N ratios were above 1 in most of the global grids, indicating that the pseudo-observation is distinguishable from the natural internal variability. Also, in some parts of the world (e.g., Canada and northern Asia, South Africa, eastern South America, eastern Australia), the S/N ratios were significant at $p = 0.05$ (i.e., greater than 1.96). These results support the original conclusions that anthropogenic forcings significantly impacted SSI. We did not add this analysis into the manuscript because Reviewer #3 suggested that the original manuscript already had too much information.

Figure E. The grid-scale S/N ratios of the 0–10 cm layer 3-month SSI of the pseudo-observation (Mean NonCMIP). The hatched areas are where the pseudo-observation S/N ratios were outside the 95% CI of the CMIP6 ALL simulations.

Figure F. The grid-scale S/N ratios of the 0–10 cm layer 3-month SSI of the pseudo-observation (Mean NonCMIP). The hatched areas are where the pseudo-observation S/N ratios were outside the 95% CI of the CMIP6 ALL simulations.

Regarding physical justification of the emergent constraint, we claim physical justifications as follows, partially based on our new analysis on the contribution of precipitation, temperature, LAI, and snowmelt to the SSI trends:

- (1) There is demonstrated emergent constraint between the historical and future trends in surface air temperature (Tokarska et al., 2020) and between the historical trends in leaf area index (LAI) and future changes in gross primary productivity, which is linearly correlated with LAI until the latter reaches saturation (Winkler et al., 2019). Temperature is the most important contributor to the SSI changes found in our study (see Fig. S9 of the revised manuscript). No emergent constraint relationship is known between historical and future trends in global precipitation or snow water equivalent. However, precipitation changes in convergence regions are closely related to temperature changes through hydrological sensitivity (Su et al., 2017), and the Northern Hemisphere spring snow cover is linearly correlated with global surface air temperature (Mudryk et al., 2020).
- (2) The spatial patterns in the correlation between the drivers of SSI and the SSI were consistent between historical and future time periods, although the magnitudes change somewhat (see Fig. S5 of the revised manuscript). Therefore, the historical spatial patterns of SSI trends should be correlated with the future spatial patterns of SSI trends through the relationship between the trends in the historical and future drivers. The magnitudes change is not an issue for emergent constraint because it can be accounted for by varying the regression coefficient over the future periods.
- (3) Because the S/N ratios of the SSI trends are indicative of spatial patterns of trends through the fingerprint (see details of calculation in Section 2 of the Supplementary Material), emergent constraint relationships should exist between the historical and future S/N ratios of the SSI.

We added the above rationale to lines 212–219 (tracked change version lines 436–443) of the revised manuscript.

Regarding the statistical significance issue, in our original results, multiple emergent constraint regressions were performed, with one individual regression for one individual month and future time period (e.g., 2001–2046, 2002–2047, ..., 2055–2100). Some of these regressions were significant and some of them not. There may be several causes of this difference in significance. First, the relative importance of the drivers to SSI trends were different from month to month (see Figs. S5 and S9 of the revised manuscript). Because the degree to which emergent constraint exists in the different drivers also differs (point 1 above), the drivers' difference will propagate to the degree to which emergent constraint exists in the SSI. Second, the uncertainty in climate projections generally increase further into the future. This is consistent with and may explain the fact that the original emergent constraint regressions were generally less significant further into the future. A recent study found that the existence of emergent constraint can vary with the degree to which the climate has changed because linearity assumptions underlying the emergent constraint regression can break down at high levels of climate change (Winkler et al., 2019). This phenomenon may be applicable to the case of our original results. Overall, the significance of the emergent constraint in the original paper should be interpreted as being seasonally and temporally dependent.

Given these discussion and caveats, we acknowledge that the reported results of the original emergent constraint can be confusing to readers. The significance of the emergent constraint was likely underestimated in the previous analysis because we did not leverage the fact that the S/N ratios were trends (i.e., the S/N ratios of two adjacent time periods were derived from a large amount of overlapping data points). As a result of this overlap, fitting one linear regression per future time period would have resulted in more parameters than necessary and reduced the statistical significance of the parameters. Therefore, in the revised paper, we reduced the number of parameters in the emergent constraint by applying a generalized additive model (GAM) (Wood, 2017), which takes the form

$$y = \beta_1 + s(x, t) + \varepsilon ,$$

where β_1 is the intercept, ε is the fitting residual, y is the future modeled S/N ratio, x is the historical modeled S/N ratio, t is the year, and $s(\cdot)$ is a tensor product smooth over x and t – which, intuitively, is a sum of the products between all pairs of the spline basis of x and the spline basis of t (Wood, 2006). This form of regression allows the relationship between y and x to change smoothly over time as a function of t . In our implementation, we used a simple linear function as the marginal smooth of x . Thus, the relationship between x and y is linear (i.e., in the same form as the linear regressions performed by past emergent constraint studies). We used a cubic regression spline as the marginal smooth of t . The new GAMs for all the months and soil layers had statistically significant term $s(\cdot)$ and achieved better fit than the original rolling regressions, as indicated by the smaller standard deviations of the fitting residuals (Figure G). Therefore, we deemed the new GAM-based emergent constraint procedure to be reasonable and has the benefit of eliminating the sometimes significant, sometimes insignificant confusion of the original rolling regressions. The constrained future S/N ratios were slightly different from the original linear constrained results, but the overall trends were similar. The results based on the new GAM emergent constraint are reported in Section 2.3 of the revised manuscript (lines 205–242 [tracked change version 407–472]). The Methods section was also revised accordingly (lines 604–619 [tracked change version 1143–1158]).

Figure G. Monthly differences in the standard deviations of the fitted residuals between the rolling linear regression in the original manuscript and the generalized additive model in the revised manuscript (the former minus the latter).

4) Presentation: The main text is relatively easy to understand, though there are so conceptual errors in describing results (e.g. detection at the 95% confidence interval etc.). But the Supplementary Information is impossible to follow. I read for more than 5 times and I am still not sure I fully understand how the authors did the analyses (I am sorry if I have misunderstood your work in my review).

Response: We apologize for the confusion. As the reviewer can see from the Track Changes version, we have cleared up the writing of both the main text and the Supplementary Material.

Reviewer #3 (Remarks to the Author):

From my first reading of the manuscript, I feel that this manuscript is very interesting but currently not suited for publication in Nature Communications. The manuscript is well written and I see the importance but the current form of the manuscript needs more than major revisions in it's current form. The topic of D&A of soil moisture trends is very relevant and interesting, certainly with the CMIP6 models included. I feel the analysis is relevant but need further improvement. Please find some suggestions below.

I find the manuscript hard to read, it is difficult to follow the different steps, approaches, and experiments. The manuscript would benefit from a smaller set of experiments or a better

description of the different experiments in a concise manner. The current manuscript balances on a lot of different analysis, different angles and it is very hard to detangle the main message. I find it very tough to get through the current form of the manuscript. I know and see that there is a lot of interesting results in the manuscript, but the current content load of the manuscript is way too high. My recommendation would be to focus on 2/3 main message to better show the reader the added value.

Response: We apologize for the confusion. As the reviewer can see from the Track Changes version, we have cleared up the writing of both the main text and the Supplementary Material. We also have made substantial changes based on the suggestions/comments from the other two reviewers in the revised version.

Are these trends significant, please provide uncertainty bounds? If I read the caption of Figure 1, they are not outside of the 95% CI. Please also clarify left and right orientated hatching, better to use hatching and some horizontal lines. I left-orientated from left to right or right to left?

Response: Thank you for these suggestions. We have denoted statistical significance of the displayed trends of the pseudo-observation (panels c, f) in Fig. 1 by applying a white mask on the insignificant areas and using contours to delineate boundary of the white-masked areas. For the trends of the ALL simulations (panels d, h) in Fig. 1, we denoted where less than 50% of the simulations had significant trends, also by white masks. We also changed the hatching to use vertical and horizontal lines.

Line 147-148 is not very satisfying, if the authors indicate potential model deficiencies it would be better to show that this is the case instead of leaving the reader without any proof.

Response: Thank you for this comment. We clarified the causes of model deficiencies by using the ratio of precipitation to potential evapotranspiration. We also showed that part of the disagreement may be caused by deficiencies in some of the reanalysis data that went into the merged soil moisture products used in this study. These revisions are reflected in lines 129–141 (tracked change version lines 210–238) and the Supplementary Material Figs. S2 and S3.

Line 151-153, another speculation that is not really support by any evidence. I would prefer

Response: Thank you for this comment. This issue is addressed in the same lines and Supplementary Material figures as the comment above.

Mention how stomate closure response affect potential evaporation rates in the future

Response: Thank you for this comment. This is implied in Discussion lines 263–267 (tracked change version lines 493–497) “Also, reduced stomatal conductance in response to drought and the future increase in atmospheric carbon dioxide concentration⁵³ mitigates the impact of rising temperatures on transpiration, which affects both the surface and root-zone soil moisture, but no such mitigation effect exists for evaporation, which mainly affects the surface layer.”

I feel that 22 extra figures in the appendix and 12 more table do not improve the readability of the manuscript. I feel that the authors should be able to provide the main message of the manuscript with a significant lower number of figures in the appendix.

Response: Thank you for this comment. We have cleared up the writing of both the main text and the Supplementary Material in the revision, and taken out various figures and tables.

Line 170, dot is at the wrong spot before the brackets

Response: Thank you for this comment. This line no longer exists in the revised manuscript because we think the information is unnecessary for the key message.

References

Berg, A., Sheffield, J., and Milly, P. C. D.: Divergent surface and total soil moisture projections under global warming, *Geophys. Res. Lett.*, 44, 236–244, <https://doi.org/10.1002/2016GL071921>, 2017.

Bindoff, N. L., Stott, P. A., AchutaRao, K. M., Allen, M. R., Gillet, N., Gutzler, D., Hansingo, K., Hergerl, G., Hu, Y., Jain, S., Mokhov, I. I., Overland, J., Perlwitz, J., Sebbari, R., and Zhang, X.: Detection and attribution of climate change: From global to regional, in: *Climate Change 2013: The Physical Science Basis. Contribution of Working Group I to the Fifth Assessment Report of the Intergovernmental Panel on Climate Change*, edited by: Stocker, T. F., Qin, D., Plattner, G.-K., Tignor, M., Allen, S. K., Boschung, J., Nauels, A., Xia, Y., Bex, V., and Midgley, P. M., Cambridge University Press, Cambridge, United Kingdom and New York, NY, USA, 867–952, <https://doi.org/10.1017/CBO9781107415324.022>, 2013.

Bonfils, C., Anderson, G., Santer, B. D., Phillips, T. J., Taylor, K. E., Cuntz, M., Zelinka, M. D., Marvel, K., Cook, B. I., Cvijanovic, I., and Durack, P. J.: Competing Influences of Anthropogenic Warming, ENSO, and Plant Physiology on Future Terrestrial Aridity, 30, 6883–6904, <https://doi.org/10.1175/JCLI-D-17-0005.1>, 2017.

Bonfils, C. J. W., Santer, B. D., Fyfe, J. C., Marvel, K., Phillips, T. J., and Zimmerman, S. R. H.: Human influence on joint changes in temperature, rainfall and continental aridity, *Nat. Clim. Chang.*, 10, 726–731, <https://doi.org/10.1038/s41558-020-0821-1>, 2020.

Cheng, S., Huang, J., Ji, F., and Lin, L.: Uncertainties of soil moisture in historical simulations and future projections, *J. Geophys. Res. Atmos.*, 122, 2239–2253, <https://doi.org/10.1002/2016JD025871>, 2017.

Cook, B. I., Smerdon, J. E., Seager, R., and Coats, S.: Global Warming and 21st Century Drying, 43, 2607–2627, <https://doi.org/10.1007/s00382-014-2075-y>, 2014.

Dai, A., Zhao, T., and Chen, J.: Climate Change and Drought: a Precipitation and Evaporation Perspective, 4, 301–312, <https://doi.org/10.1007/s40641-018-0101-6>, 2018.

- Deng, Y., Wang, S., Bai, X., Luo, G., Wu, L., Cao, Y., Li, H., Li, C., Yang, Y., Hu, Z., and Tian, S.: Variation Trend of Global Soil Moisture and Its Cause Analysis, 110, 105939, <https://doi.org/10.1016/j.ecolind.2019.105939>, 2020.
- Douville, H. and Plazzotta, M.: Midlatitude Summer Drying: An Underestimated Threat in CMIP5 Models?, 44, 9967–9975, <https://doi.org/10.1002/2017GL075353>, 2017.
- Douville, H., Ribes, A., Decharme, B., Alkama, R., and Sheffield, J.: Anthropogenic Influence on Multidecadal Changes in Reconstructed Global Evapotranspiration, 3, 59–62, <https://doi.org/10.1038/nclimate1632>, 2013.
- Fan, X., Duan, Q., Shen, C., Wu, Y., and Xing, C.: Global surface air temperatures in CMIP6: historical performance and future changes, *Environ. Res. Lett.*, 15, 104056, <https://doi.org/10.1088/1748-9326/abb051>, 2020.
- Feng, S. and Fu, Q.: Expansion of Global Drylands Under a Warming Climate, 13, 10081–10094, <https://doi.org/10.5194/acp-13-10081-2013>, 2013.
- Field, C. B., Barros, V., Stocker, T. F., and Dahe, Q. (Eds.): Managing the risks of extreme events and disasters to advance climate change adaptation: Special report of the intergovernmental panel on climate change, Cambridge University Press, Cambridge, <https://doi.org/10.1017/CBO9781139177245>, 2012.
- Fu, Y. H., Zhou, X., Li, X., Zhang, Y., Geng, X., Hao, F., Zhang, X., Hanninen, H., Guo, Y., and De Boeck, H. J.: Decreasing control of precipitation on grassland spring phenology in temperate China, *Global Ecol. Biogeogr.*, 30, 490–499, <https://doi.org/10.1111/geb.13234>, 2021.
- Gevaert, A. I., Miralles, D. G., Jeu, R. A. M., Schellekens, J., and Dolman, A. J.: Soil moisture-temperature coupling in a set of land surface models, *J. Geophys. Res. Atmos.*, 123, 1481–1498, <https://doi.org/10.1002/2017JD027346>, 2018.
- Greve, P. and Seneviratne, S. I.: Assessment of Future Changes in Water Availability and Aridity, 42, 5493–5499, <https://doi.org/10.1002/2015GL064127>, 2015.
- Hu, Y., Huang, H., and Zhou, C.: Widening and weakening of the Hadley circulation under global warming, *Sci.*, 63, 640–644, <https://doi.org/10.1016/j.scib.2018.04.020>, 2018.
- Lawrence, D. M., Hurtt, G. C., Arneth, A., Brovkin, V., Calvin, K. V., Jones, A. D., Jones, C. D., Lawrence, P. J., de Noblet-Ducoudré, N., Pongratz, J., Seneviratne, S. I., and Shevliakova, E.: The Land Use Model Intercomparison Project (LUMIP) contribution to CMIP6: rationale and experimental design, *Geosci. Model Dev.*, 9, 2973–2998, <https://doi.org/10.5194/gmd-9-2973-2016>, 2016.
- Li, W., Migliavacca, M., Forkel, M., Walther, S., Reichstein, M., and Orth, R.: Revisiting global vegetation controls using multi-layer soil moisture, <https://doi.org/10.1002/essoar.10504463.1>, 2020.

Lian, X., Piao, S., Chen, A., Huntingford, C., Fu, B., Li, L. Z. X., Huang, J., Sheffield, J., Berg, A. M., Keenan, T. F., McVicar, T. R., Wada, Y., Wang, X., Wang, T., Yang, Y., and Roderick, M. L.: Multifaceted Characteristics of Dryland Aridity Changes in a Warming World (Accepted), n.a., n.a., <https://doi.org/10.1038/s43017-021-00144-0>, 2021.

Liu, L., Gudmundsson, L., Hauser, M., Qin, D., Li, S., and Seneviratne, S. I.: Soil moisture dominates dryness stress on ecosystem production globally, *Nat Commun*, 11, 4892, <https://doi.org/10.1038/s41467-020-18631-1>, 2020.

Liu, Y., Liu, Y., and Wang, W.: Inter-Comparison of Satellite-Retrieved and Global Land Data Assimilation System-Simulated Soil Moisture Datasets for Global Drought Analysis, 220, 1–18, <https://doi.org/10.1016/j.rse.2018.10.026>, 2019.

Lu, J., Carbone, G. J., and Grego, J. M.: Uncertainty and Hotspots in 21st Century Projections of Agricultural Drought from CMIP5 Models, 9, 4922, <https://doi.org/10.1038/s41598-019-41196-z>, 2019.

Martens, B., Miralles, D. G., Lievens, H., van der Schalie, R., de Jeu, R. A. M., Fernández-Prieto, D., Beck, H. E., Dorigo, W. A., and Verhoest, N. E. C.: GLEAM v3: satellite-based land evaporation and root-zone soil moisture, *Geosci. Model Dev.*, 10, 1903–1925, <https://doi.org/10.5194/gmd-10-1903-2017>, 2017.

Marvel, K. and Bonfils, C.: Identifying External Influences on Global Precipitation, 110, 19301–19306, <https://doi.org/10.1073/pnas.1314382110>, 2013.

Mudryk, L., Santolaria-Otín, M., Krinner, G., Ménégoz, M., Derksen, C., Brutel-Vuilmet, C., Brady, M., and Essery, R.: Historical Northern Hemisphere snow cover trends and projected changes in the CMIP6 multi-model ensemble, *The Cryosphere*, 14, 2495–2514, <https://doi.org/10.5194/tc-14-2495-2020>, 2020.

O, S. and Orth, R.: Global soil moisture data derived through machine learning trained with in-situ measurements, *Sci. Data*, 8, 170, <https://doi.org/10.1038/s41597-021-00964-1>, 2021.

Santer, B. D., Po-Chedley, S., Zelinka, M. D., Cvijanovic, I., Bonfils, C., Durack, P. J., Fu, Q., Kiehl, J., Mears, C., Painter, J., Pallotta, G., Solomon, S., Wentz, F. J., and Zou, C.-Z.: Human influence on the seasonal cycle of tropospheric temperature, *Science*, 361, eaas8806, <https://doi.org/10.1126/science.aas8806>, 2018.

Sarojini, B. B., Stott, P. A., and Black, E.: Detection and Attribution of Human Influence on Regional Precipitation, 6, 669–675, <https://doi.org/10.1038/nclimate2976>, 2016.

Schlaepfer, D. R., Bradford, J. B., Lauenroth, W. K., Munson, S. M., Tietjen, B., Hall, S. A., Wilson, S. D., Duniway, M. C., Jia, G., Pyke, D. A., Lkhagva, A., and Jamiyansharav, K.: Climate Change Reduces Extent of Temperate Drylands and Intensifies Drought in Deep Soils, 8, 14196, <https://doi.org/10.1038/ncomms14196>, 2017.

- Su, C.-H., Ryu, D., Dorigo, W., Zwieback, S., Gruber, A., Albergel, C., Reichle, R. H., and Wagner, W.: Homogeneity of a global multisatellite soil moisture climate data record, *Geophys. Res. Lett.*, 43, 11245–11252, <https://doi.org/10.1002/2016GL070458>, 2016.
- Su, H., Jiang, J. H., Neelin, J. D., Shen, T. J., Zhai, C., Yue, Q., Wang, Z., Huang, L., Choi, Y.-S., Stephens, G. L., and Yung, Y. L.: Tightening of tropical ascent and high clouds key to precipitation change in a warmer climate, *Nat. Commun.*, 8, 15771, <https://doi.org/10.1038/ncomms15771>, 2017.
- Swann, A. L. S., Hoffman, F. M., Koven, C. D., and Randerson, J. T.: Plant responses to increasing CO₂ reduce estimates of climate impacts on drought severity, *Proc. Natl. Acad. Sci. USA*, 113, 10019–10024, <https://doi.org/10.1073/pnas.1604581113>, 2016.
- Tobin, K. J., Torres, R., Crow, W. T., and Bennett, M. E.: Multi-decadal analysis of root-zone soil moisture applying the exponential filter across CONUS, *Hydrol. Earth Syst. Sci.*, 21, 4403–4417, <https://doi.org/10.5194/hess-21-4403-2017>, 2017.
- Tokarska, K. B., Stolpe, M. B., Sippel, S., Fischer, E. M., Smith, C. J., Lehner, F., and Knutti, R.: Past warming trend constrains future warming in CMIP6 models, *Sci. Adv.*, 6, eaaz9549, <https://doi.org/10.1126/sciadv.aaz9549>, 2020.
- Wang, Y., Mao, J., Hoffman, F., Jin, M., Shi, X., Wullschleger, S., and Dai, Y.: Development of observation-based global multi-layer soil moisture products for the period 1970-2016, *Earth Syst. Sci. Data*, Accepted, <https://doi.org/10.5194/essd-2021-84>, 2021.
- Winkler, A. J., Myneni, R. B., and Brovkin, V.: Investigating the applicability of emergent constraints, *Earth Syst. Dynam.*, 10, 501–523, <https://doi.org/10.5194/esd-10-501-2019>, 2019.
- Wood, S. N.: Low-rank scale-invariant tensor product smooths for generalized additive mixed models, 62, 1025–1036, <https://doi.org/10.1111/j.1541-0420.2006.00574.x>, 2006.
- Wood, S. N.: *Generalized Additive Models: an introduction with R*, 2nd ed., Chapman and Hall/CRC, 2017.
- Xia, Y., Hu, Y., and Liu, J.: Comparison of trends in the Hadley circulation between CMIP6 and CMIP5, *Sci.*, 65, 1667–1674, <https://doi.org/10.1016/j.scib.2020.06.011>, 2020.
- Xu, C., McDowell, N. G., Fisher, R. A., Wei, L., Sevanto, S., Christoffersen, B. O., Weng, E., and Middleton, R. S.: Increasing Impacts of Extreme Droughts on Vegetation Productivity Under Climate Change, 9, 948–953, <https://doi.org/10.1038/s41558-019-0630-6>, 2019.
- Xu, L., Chen, N., Zhang, X., Moradkhani, H., Zhang, C., and Hu, C.: In-situ and triple-collocation based evaluations of eight global root zone soil moisture products, *Remote Sens. Environ.*, 254, 112248, <https://doi.org/10.1016/j.rse.2020.112248>, 2021.
- Yang, T., Ding, J., Liu, D., Wang, X., and Wang, T.: Combined use of multiple drought indices for global assessment of dry gets drier and wet gets wetter paradigm, *J. Climate*, 32, 737–748, <https://doi.org/10.1175/JCLI-D-18-0261.1>, 2019.

Zhang, X., Zwiers, F. W., Hegerl, G. C., Lambert, F. H., Gillett, N. P., Solomon, S., Stott, P. A., and Nozawa, T.: Detection of Human Influence on Twentieth-Century Precipitation Trends, 448, 461–465, <https://doi.org/10.1038/nature06025>, 2007.

REVIEWER COMMENTS

Reviewer #1 (Remarks to the Author):

After reading the revised draft and the authors' responses, I believe the revised draft has sufficiently addressed all of my previous concerns and therefore recommend acceptance of the article. I appreciate the authors' efforts in adding graphs to support their discussions on model-data disagreements and the forcing mechanisms. The readability of the draft was much better than the last one. The choice of GAM in the revised application of emergent constraint is different from traditional linear regression methods, but appeared to be justifiable based on the improved residuals. I think future studies that want to apply emergent constraint on rolling future periods and future detection and attribution studies can benefit from knowledge of such a new procedure.

Reviewer #2 (Remarks to the Author):

I appreciate significant efforts the authors made in addressing my comments. I am convinced there are added values to look into different soil layers and soil moistures. I am also convinced the merged soil moisture data is potentially useful for the purpose of this paper if its caveat is carefully considered when interpreting the analyses. But I am not convinced the detection and attribution method is implemented properly and in particular the interpretation of the relevant analyses is correct. I also don't think what the authors consider to be innovative emergent constraint adds any value.

1) Detection and attribution:

a. Overall, there may be some useful information that ALL could be detected in SSI during certain months but the description and interpretation are confusion and may contain significant errors.

b. Method: I am not sure if the authors fully understand the statistical concepts used in the analysis. Their response does not give the impression that they do. Stating that the 5% significance level and the 95% confidence interval are the same thing of different terminologies is a clear example. The lack of proper understanding in the statistical concepts makes it difficult for the authors to properly describe and interpret what the analyses tell them. Their statement (lines 183-187) "also, the signals of the pseudo-observation ... partially attributable to the AER forcing". Every panel in Fig 3 shows very clearly that AER distribution is almost the same as that of noise. Thus any claim that AER can be detected in the observation, and more importantly the observed changes can be attributed to (even in part), cannot be

correct, it does not matter if the observed s/n is within or outside the 95% range of AER. The distributions of ALL, ANT, and GHG overlap with that of noise but they are also quite away from the central of the noise distribution. One may be able to argue that if the observed S/N is above of the 95th percentile of the noise distribution and if that is also within certain range of ALL/ANT/GHG, one could make a detection claim. But to do so, one needs to clearly define what the threshold is to make such a claim to avoid misinterpretation.

c. Interpretation: in addition to problems mentioned in a), there are also issues in the meaning of the fingerprints, in particular the lack of physical understanding/interpretation of fingerprints. As described in page ... and shown in fig 2, the fingerprints for AER and GHG are very similar for many months and latitudes. But what are the physics for AER and GHG to behave similarly (wet or dry for the same latitude and during the same months)? What is the implication of this in the interpretation of attribution claim?

2) Emergent constraint:

a. Overall, the emergent constraint is confusion and does not seem to add any value.

b. The description of the method is confusion and does not seem to be consistent with the description of the results. Lines 604-608 indicate the regression was done between modelled future periods and modelled historical period, but lines 211-213 suggest to relate to the (observed) historical ratio to correct model bias. If the regression is meant to correct model biases, one would expect the future ratio to regress on to observed rather than the modelled ratio.

c. While one may use the observed ratio to correct modelled ratio for the same historical period, one cannot simply extrapolate that adjustment factor to the future period without making explicit assumption such as same warming rate or soil moisture response rate because the s/n ratio is computed for the same fixed 46 year period. But regressing modelled future period's ratio to the modelled historical ratio does not make sense at all. If the signal is sufficiently strong and if the warming rate is consistent along time (assuming soil moisture respond proportionally to warming rate) one would expect the same s/n ratio from the same model, any difference is simply noise/uncertainty. On the other hand, if future warming rates differ from the past (which is unlikely the case under the SSP8.5), the difference may also be affected by future warming rate which shall not be corrected.

Reviewer #1 (Remarks to the Author):

After reading the revised draft and the authors' responses, I believe the revised draft has sufficiently addressed all of my previous concerns and therefore recommend acceptance of the article. I appreciate the authors' efforts in adding graphs to support their discussions on model-data disagreements and the forcing mechanisms. The readability of the draft was much better than the last one. The choice of GAM in the revised application of emergent constraint is different from traditional linear regression methods, but appeared to be justifiable based on the improved residuals. I think future studies that want to apply emergent constraint on rolling future periods and future detection and attribution studies can benefit from knowledge of such a new procedure.

Reviewer #2 (Remarks to the Author):

I appreciate significant efforts the authors made in addressing my comments. I am convinced there are added values to look into different soil layers and soil moistures. I am also convinced the merged soil moisture data is potentially useful for the purpose of this paper if it's caveat is carefully considered when interpreting the analyses. But I am not convinced the detection and attribution method is implemented properly and in particular the interpretation of the relevant analyses is correct. I also don't think what the authors consider to be innovative emergent constraint adds any value.

1) Detection and attribution:

a. Overall, there may be some useful information that ALL could be detected in SSI during certain months but the description and interpretation are confusion and may contain significant errors.

b. Method: I am not sure if the authors fully understand the statistical concepts used in the analysis. Their response does not give the impression that they do. Stating that the 5% significance level and the 95% confidence interval are the same thing of different terminologies is a clear example. The lack of proper understanding in the statistical concepts makes it difficult for the authors to properly describe and interpret what the analyses tell them. Their statement (lines 183-187) "also, the signals of the pseudo-observation ... partially attributable to the AER forcing". Every panel in Fig 3 shows very clearly that AER distribution is almost the same as that of noise. Thus any claim that AER can be detected in the observation, and more importantly the observed changes can be attributed to (even in part), cannot be correct, it does not matter if the observed s/n is within or outside the 95% range of AER. The distributions of ALL, ANT, and GHG overlap with that of noise

but they are also quite away from the central of the noise distribution. One may be able to argue that if the observed S/N is above of the 95th percentile of the noise distribution and if that is also within certain range of ALL/ANT/GHG, one could make a detection claim. But to do so, one needs to clearly define what the threshold is to make such a claim to avoid misinterpretation.

Thank you for pointing out the potential errors in the methodology. Point-by-point responses are below:

- i. Regarding the 5% significance level and 95% confidence interval terminology

In the previous revision, we actually replaced “95% confidence level” with “ $p = 0.05$ ”. In the previous round of comments, the reviewer said that “this paper would state to have detected at the 95% confidence interval”. Since we only ever stated detectability at “95% confidence level”, we believed “confidence level” was what the reviewer meant, but used the “95% confidence level/interval” wording in the response letter in order to echo the reviewer’s wording. We are sorry that it has caused further confusion.

We did state in the response letter that “when a test statistic is within the 95% CI, there is no statistical significance; when outside, there is statistical significance at a 5% level (total probability is always 100%, hence $100\% - 95\% = 5\%$)”. This we believe to be correct. For example, to quote a 2016 paper on statistical concepts¹: “The effect sizes whose test produced $P > 0.05$ will typically define a range of sizes ... that would be considered more compatible with the data ... This range corresponds to a $1 - 0.05 = 0.95$ or 95% confidence interval.”

ii. Regarding the similarity between the AER and the noise distributions

Thank you for pointing this out. This is indeed a problem that we failed to recognize. In the revised manuscript, we added the following to lines XXX-XXX of the main text:

“However, the 95% CIs of the AER signals were very similar to the noises, suggesting that the effects of AER on SSI changes are too small and uncertain to warrant attribution.”

However, we think the potential existence of AER contributions to the signals cannot be completely ignored. In Fig. 3, the significant overlap between the 95% CI of the AER forcing and the noise suggested that the AER forcing has no effect on SSI during our period of interest. But the AER signals and the noises in Fig. 3 were calculated by projecting the simulations on the ALL fingerprints, which mainly reflects the GHG effects. When the AER signals and the noises were projected onto the AER fingerprints, the separation between the AER signals and the noises was improved, especially when the pseudo-observation signals were detected (Figure A1i-k, u-w). We added lines 209-211 (lines 213-215 in the tracked version) to the main text to reflect this difference. Furthermore, at least two previous studies that used different detection and attribution (D&A) methods suggested potential contributions by the AER forcing to drought or soil moisture^{2,3}. Current D&A of AER effects on hydrological changes is still impaired by the spatiotemporal complexity of aerosol forcings and model inadequacies in aerosol indirect effects⁴. Therefore, we think it is beneficial to report the potential AER contributions along with the caveats in order to promote future studies in this area.

Figure A1. Signals of the pseudo-observation (Mean NonCMIP), the shapes and 95% CIs of the Gaussian distributions fitted to the signals of the AER simulations, and the 95% and 99% CIs of natural internal variability (noise). All the signals were for 1971–2016 and the noises were for the window length of 46 years. All the signals and noises were projected on the AER fingerprints.

c. Interpretation: in addition to problems mentioned in a), there are also issues in the meaning of the fingerprints, in particular the lack of physical understanding/interpretation of fingerprints. As described in page ... and shown in fig 2, the fingerprints for AER and GHG are very similar for

many months and latitudes. But what are the physics for AER and GHG to behave similarly (wet or dry for the same latitude and during the same months)? What is the implication of this in the interpretation of attribution claim?

Thank you for these questions. The fingerprints in our D&A method can be interpreted as the large-scale spatial signatures caused by the forcings⁵. We added the following to lines 143-146 (the same lines in the tracked version) to the main text:

“In pattern-based D&A analysis, the fingerprint represents the large-scale spatial signature caused by the forcings, and higher S/N ratios means greater tendency of the spatiotemporal field of interest (here the SSI) to become similar to the fingerprint over time .”

The SSI fingerprints were usually similar to the SSI trends, and we intended the discussion on the mechanisms of the SSI trends to be also applicable to the SSI fingerprints. We edited lines 154-169 (154-171 in the tracked version) and line 298-317 (lines 307-333) of the main text to better reflect this intention.

The AER and GHG fingerprints show similarities in Jan-May in the northern mid-latitudes, and Jun-Dec in the northern high-latitudes and southern low- and mid-latitudes. The Jan-Jun AER fingerprints of the surface soil layer, and Apr-Jun AER fingerprints of the root-zone soil layer are not associated with statistically significant trends in the principal components (manuscript SI Fig. S4). Therefore, the AER fingerprints of these months and soil layers do not reflect directional changes in the SSI pattern like the GHG fingerprints. We added lines 161-163 (lines 162-165 in the tracked version) to the main text to note this. The Jul-Dec AER fingerprints of the surface layer and the Jul-Mar fingerprints of the root-zone layer are associated with significant trends in the principal components (manuscript SI Fig. S4). Still, their similarities to the GHG fingerprints should be accidental and caused by different mechanisms (see next paragraph). Also, any similarity between the GHG and AER fingerprints is not a problem for the D&A, because the D&A involves using the entire latitudinal pattern, which is always different between GHG and AER.

Figure A3 and Figure A4 show that the latitudinal trends in the drivers of soil moisture (precipitation, temperature, leaf area index, snow water equivalent) are very different between the GHG and AER simulations. In the GHG simulations, the precipitation trends are generally positive in the Northern Hemisphere, which would induce SSI wetting trends (Figure A3). Therefore, the widespread SSI drying trends must be driven by the increasing evapotranspiration caused by increasing temperature and leaf area index, and decreasing snow water equivalent that exposes more soil to evaporation (Figure A3). In the Southern Hemisphere, increasing temperature and leaf area index, and decreasing precipitation both contributed to drying SSI in the GHG simulations (Figure A3). In contrast, in the AER simulations, temperature is only increasing above 30°N and decreasing elsewhere, and the AER-forced trends in the drivers of SSI are mostly weaker than in the GHG simulations (Figure A4). The increasing temperatures above 30°N is centered in Europe (Figure A4), which is opposite to the effects of aerosol loading in Europe⁴ and consistent with decreasing sulfate emissions in Europe after 1970⁶. The small patch of drying in the AER fingerprint between 0° and 20°N (Fig. 2) corresponds to a band of decreasing precipitation (Figure A3) that is centered in East Asia (Figure A4). This is consistent with the increasing aerosol emissions in Asia after 1970 and the weakening effect of aerosols on monsoon circulation^{4,6,7}. The drying in the AER fingerprint in the Southern Hemisphere (Fig. 2) corresponds to decreasing precipitation in the

Amazon and Brazilian Cerrado regions and the Southeast Asian islands (Figure A4). The former two are consistent with previously simulated effects of aerosols reductions in the US and Europe⁸. The last region is subject to the influence of monsoon circulation like East Asia. We edited lines 308-317 (lines 318-333 in the tracked version) in the main text to briefly reflect these mechanisms.

Figure A2. Trends in latitudinally averaged precipitation (pr), air temperature (tas), leaf area index (lai), and snow water equivalent (snw) over 1971-2016 in the GHG simulations. The trends were calculated using least squares over each simulation, averaged over the ensemble members of each model, and then averaged over the models. Vertical stippling indicates where more than 90% of the models agreed with the displayed sign, and horizontal stippling indicates more than 80%.

Figure A3. Trends in precipitation (pr), air temperature (tas), leaf area index (lai), and snow water equivalent (snw) over 1971-2016 in the AER simulations. The trends were calculated using least squares over each simulation, averaged over the ensemble members of each model, and then averaged over the models. Vertical stippling indicates where more than 90% of the models agreed with the displayed sign, and horizontal stippling indicates more than 80%.

Figure A4. Trends in precipitation (pr), air temperature (tas), leaf area index (lai), and snow water equivalent (snw) over 1971-2016 in the AER simulations. The trends were calculated using least squares over each simulation, averaged over the ensemble members of each model, and then averaged over the models. Vertical stippling indicates where more than 90% of the models agreed with the displayed sign, and horizontal stippling indicates more than 80%.

2) Emergent constraint:

a. Overall, the emergent constraint is confusion and does not seem to add any value.

The value of emergent constraint in this study lies in giving better estimate of the future projected S/N ratios than the simple median or unweighted average over the models. Without

applying the emergent constraint, the median modeled S/N ratios often have considerable positive biases compared to the observed S/N ratios (compare the median of the ALL distributions to the observation in Fig. 3, and the dashed versus solid lines in Fig. 4). The modeled S/N ratios are probably too high, because the modeled signals and noises have a common source of error, i.e., the shared model physics. The observed S/N ratios use observation-based signals but modeled noises, and thus do not have this common error. By constraining the future modeled S/N ratios, we avoid drawing too strong conclusions about the strength of future S/N ratios. We edited lines 220-227 (225-232 in the traced version) in the main text to better reflect this rationale.

b. The description of the method is confusion and does not seem to be consistent with the description of the results. Lines 604-608 indicate the regression was done between modelled future periods and modelled historical period, but lines 211-213 suggest to relate to the (observed) historical ratio to correct model bias. If the regression is meant to correct model biases, one would expect the future ratio to regress on to observed rather than the modelled ratio.

We indeed performed the regression between historical and future modeled S/N ratios by treating the historical-future pair of each individual model as an x-y pair in the regression. Then, we plugged the observed historical ratio as an x-value into the same regression to generate point estimates of y-values (“constrained future S/N ratios”), as well as corresponding uncertainty intervals. This is standard practice in emergent constraint, which is a well-established method to reduce the uncertainty in future climate projections⁹⁻¹¹. We edited lines 227-232 (lines 232-238 in the tracked version) in the main text to better describe the emergent constraint method, which is also described in the Methods section.

The emergent constraint method is therefore different from traditional bias-correction, which extrapolates the historical biases of models to correct the future projections of the same models. As noted by the reviewer in point c, this extrapolation makes unrealistic assumptions. Emergent constraint makes different assumptions, which we detail in the response to point c. To avoid confusing the readers with bias-correction, we paraphrased all the bias-correction wordings using “constrained” or “emergent constraint” in the revised text.

c. While one may use the observed ratio to correct modelled ratio for the same historical period, one cannot simply extrapolate that adjustment factor to the future period without making explicit assumption such as same warming rate or soil moisture response rate because the s/n ratio is computed for the same fixed 46 year period. But regressing modelled future period’s ratio to the modelled historical ratio does not make sense at all. If the signal is sufficiently strong and if the warming rate is consistent along time (assuming soil moisture respond proportionally to warming rate) one would expect the same s/n ratio from the same model, any difference is simply noise/uncertainty. On the other hand, if future warming rates differ from the past (which is unlikely the case under the SSP8.5), the difference may also be affected by future warming rate which shall not be corrected.

As mentioned above, the emergent constraint method does not extrapolate any historical model-to-observed adjustment factor to the future period. Instead, emergent constraint extrapolates the historical-future relationship from the model world to the observational world.

This extrapolation requires the models to be sufficiently realistic and the relationship between the historical and future variables of interest (in our case the S/N ratios) to be physically valid. Sufficient realism of the models is required by the entire D&A process, and the issue is discussed in Section 2.1 and Section 2.2 lines 170-175 (172-177 in the tracked version) of our manuscript. Physical justification of the historical-future relationship in the S/N ratios is in lines 232-244 (lines 238-251 in the tracked version) of the main text. We edited these lines slightly to reflect the warming rate and soil moisture sensitivity discussion below.

Regarding the two situations that the reviewer describes, if the warming rate and the soil moisture's sensitivity to warming are the same for any given model between the historical and future periods, the situation works well for emergent constraint. If all the modeled S/N ratios are constant and the models are realistic, then it seems reasonable to believe that the best estimate of the yet-to-be-observed future S/N ratio is the historical observed S/N ratio. The emergent constraint will give a 1:1 regression equation between the modeled historical and future S/N ratios, and this regression equation will indeed give a constrained future S/N ratio that is equal to the historical observed S/N ratio.

If the warming rates and the soil moisture's sensitivities differ between the historical and future periods, there can still be an emergent constraint between modeled historical and future S/N ratios, if the models with higher (lower) historical warming rates and historical sensitivity continue to have higher (lower) future warming rates and future sensitivity. In this situation, it seems reasonable to believe that higher (lower) historical observed S/N ratios implies higher (lower) future to-be-observed S/N ratios, which is what emergent constraint will give. A past study already demonstrated historical-future relationship in modeled warming rates¹⁰. The historical-future relationships in the sensitivity of S/N ratios to warming rates are shown for 2025-2070 in Figure A5; other future periods have similar results. Although the linear relationships of Dec-Apr for the 0-100 cm soil layer do not have positive slopes, the constrained S/N ratios of these months are very close to, and therefore no worse than, the simple multi-model median S/N ratios (manuscript SI Fig. 4). In May-Aug when the future S/N ratios considerably exceed the historical ranges (manuscript SI Fig. 4), the positive linear relationships are always clear. These results support the validity of the use of emergent constraint in this study. We edited lines 232-244 (lines 238-251 in the tracked version) of the main text to reflect the new information in Figure A5.

Figure A5. Relationships between the historical and future sensitivities of the modeled S/N ratios to modeled warming rates (T_{trend} , $^{\circ}\text{C year}^{-1}$) in each month of the year. The historical period is 1971-2016, and the future period is 2025-2070. Each data point indicates one model, and the error bars indicate ± 1 standard deviation across the model's ensemble members. The warming rates were calculated by least squares.

References

1. Greenland, S. *et al.* Statistical tests, P values, confidence intervals, and power: a guide to misinterpretations. *Eur J Epidemiol* **31**, 337–350 (2016).
2. Marvel, K. *et al.* Twentieth-century hydroclimate changes consistent with human influence. *Nature* **569**, 59–65 (2019).
3. Qiao, L., Zuo, Z., Xiao, D. & Bu, L. Detection, attribution, and future response of global soil moisture in summer. *Frontiers in Earth Science* **9**, (2021).
4. Liu, L. *et al.* A PDRMIP multimodel study on the impacts of regional aerosol forcings on global and regional precipitation. *Journal of Climate* **31**, 4429–4447 (2018).
5. D, S. B. *et al.* Human influence on the seasonal cycle of tropospheric temperature. *Science (1979)* **361**, eaas8806 (2018).
6. Wang, Y., Jiang, J. H. & Su, H. Atmospheric responses to the redistribution of anthropogenic aerosols. *Journal of Geophysical Research: Atmospheres* **120**, 9625–9641 (2015).
7. Streets, D. G. *et al.* Anthropogenic and natural contributions to regional trends in aerosol optical depth, 1980–2006. *Journal of Geophysical Research: Atmospheres* **114**, (2009).
8. Westervelt, D. M. *et al.* Connecting regional aerosol emissions reductions to local and remote precipitation responses. *Atmos. Chem. Phys.* **18**, 12461–12475 (2018).
9. Winkler, A. J., Myrneni, R. B. & Brovkin, V. Investigating the applicability of emergent constraints. *Earth Syst. Dynam.* **10**, 501–523 (2019).
10. Tokarska, K. B. *et al.* Past warming trend constrains future warming in CMIP6 models. *Science Advances* **6**, eaaz9549 (2022).
11. Douville, H. & Plazzotta, M. Midlatitude summer drying: An underestimated threat in CMIP5 models? *Geophysical Research Letters* **44**, 9967–9975 (2017).

REVIEWER COMMENTS

Reviewer #2 (Remarks to the Author):

I appreciate authors' effort in responding to my comments and in editing the text, I am less confused now. But I must admit it is still a pain to read the paper, and I am not sure if I fully have understood what the authors did. While the technical aspects of the paper are properly ok, I don't have the confidence my assessment is correct. This is because some important details are not clearly spelled out, but some not so important extras (and in some cases perhaps even not relevant) were added. Such convoluted presentation would not help with the creditability of the paper, especially for paper that wants to be published in Nature Communication. The problem is perhaps due to careless writing or not paying enough attention to the details, or disconnect between an author who did the calculation and an author who did a writing, lack of understanding the technics by whoever holding the pen, or a combination of some of these. As the authors are still not able to pay enough attention to produce clear presentation in the second revision, I'm done with reviewing this paper and do not want to read another revision. I don't have the energy to point out every place that I feel difficult to understand, rather, I will just give a few examples to demonstrate the problems in a hope that this would give the authors a sense of the problems and perhaps a senior author who can write clearly will spend some time to carefully clean the paper.

1) The calculation of SSI was done on three time scales ($k=1,3,6$ months, line #88 of the supplement) and D&A analysis was conducted on 3-month SSI (line 148). But exactly what February-July (line #141) means here? Does it mean for all 3-month including DJF, JFM, ..., MJJ? As this is also a 6-month period, is this for the 6-month period starting F-J? There was also mentioning for detection in individual months, are they for 1-month scale of 3-month scale? Similar problem appears in many places in the paper.

2) There is a logic problem in the description of D/A results (lines #185-228). For example (line 191), the signals of pseudo-observation were detectable (meaning not within the 95% range of noise trends) ..., and later the detected signals ... were within the 95% range of the ALL or ANT signals ..." thereby the observed change is attributed to the external forcing (ALL or ANT). But there is a problem in that most of the ALL and ANT signal trends fall within the 95 range of the noise trend. So if I follow the logic, the observed signal is detected (outside of 95% range of noise trends) which is attributed to (consistent with) the response to ALL/ANT (as within the 95% range), and the latter is also consistent with noise (as most of the ALL/ANT trends are within the 95% range of noise trend). Can I then say the observed signal is also consistent with noise? I am pointing this out because there is a need to clearly define what detection means and what attribution means, and why you say ALL/ANT signals are different from noises. Without these, it is impossible to interpret your results (and your interpretation as well). Also, what do you mean by good or poor separation between a signal and noise (e.g., line 444)? There needs to be some definition to characterize.

3) The method section introduced signal estimate for the model responses that involves $Z_{(f2f1)}$ and authors indicated $f2$ and $f1$ can be the same or different. But when you described AER is not detected, did you project both observations and AER to fingerprint of ALL (in this case $f2$ is different from $f1$), or

both observations and AER to AER fingerprint, or observation to ALL fingerprint and AER to AER fingerprint?

4) I appreciate the authors' explanation regarding how they did the regression to obtain the constrained S/N ratio. I now understand what they did. But the description of this important detail cannot be found anywhere in the paper or in the supplement.

Reviewer #2 (Remarks to the Author):

I appreciate authors' effort in responding to my comments and in editing the text, I am less confused now. But I must admit it is still a pain to read the paper, and I am not sure if I fully have understood what the authors did. While the technical aspects of the paper are properly ok, I don't have the confidence my assessment is correct. This is because some important details are not clearly spelled out, but some not so important extras (and in some cases perhaps even not relevant) were added. Such convoluted presentation would not help with the creditability of the paper, especially for paper that wants to be published in Nature Communication. The problem is perhaps due to careless writing or not paying enough attention to the details, or disconnect between an author who did the calculation and an author who did a writing, lack of understanding the technics by whoever holding the pen, or a combination of some of these. As the authors are still not able to pay enough attention to produce clear presentation in the second revision, I'm done with reviewing this paper and do not want to read another revision. I don't have the energy to point out every place that I feel difficult to understand, rather, I will just give a few examples to demonstrate the problems in a hope that this would give the authors a sense of the problems and perhaps a senior author who can write clearly will spend some time to carefully clean the paper.

Thank you for pointing out the confusion in the writing. In this revision, we did not change any results, but made substantial text edits to provide more details on the D&A method and the emergent constraint method, and to improve the readability and technical accuracy of the reporting. Responses to the individual comments are below.

1) The calculation of SSI was done on three time scales ($k=1,3,6$ months, line #88 of the supplement) and D&A analysis was conducted on 3-month SSI (line 148). But exactly what February-July (line #141) means here? Does it mean for all 3-month including DJF, JFM, ..., MJJ? As this is also a 6-month period, is this for the 6-month period starting F-J? There was also mentioning for detection in individual months, are they for 1-month scale of 3-month scale? Similar problem appears in many places in the paper.

Thank the reviewer for pointing this issue out. February-July means for all 3-month including DJF, JFM, ..., MJJ. In general, the referred month is the last month in the 3 or 6-month period. This was mentioned in the previous submitted manuscript, Methods section, in a parenthesis ("... the D&A analysis, which was conducted separately for each month of the year (last month in the 3-month moving window for the SSI)", previous submitted manuscript lines 608-610). We removed these lines in the revised manuscript and mentioned this more clearly when describing the SSI (lines 607-610, tracked version 670-673):

"The calculated 3-month SSI of each month reflects the average soil moisture conditions of the current month and previous two months (e.g., the 3-month SSI in a February reflected the average soil moisture between previous year's December and this year's January and February)."

We also re-iterated this fact in the SI lines 93-99 (tracked version lines 112-118):

"For convenience, we always referred to the SSI values by the last month that the averaging periods ended in. For example, the 3-month SSI of a February would be based on the average soil moisture of the previous year's December and this year's January–February, and the 6-month SSI of a June would be based on the averaged soil moisture of this year's January–June. The same definition applied when the

SSI of multiple months were averaged (e.g., the 3-month SSI of DJF means the average value of the 3-month SSI of December, January, and February).”

2) There is a logic problem in the description of D/A results (lines #185-228). For example (line 191), the signals of pseudo-observation were detectable (meaning not within the 95% range of noise trends) ..., and later the detected signals ... were within the 95% range of the ALL or ANT signals ...” thereby the observed change is attributed to the external forcing (ALL or ANT). But there is a problem in that most of the ALL and ANT signal trends fall within the 95 range of the noise trend. So if I follow the logic, the observed signal is detected (outside of 95% range of noise trends) which is attributed to (consistent with) the response to ALL/ANT (as within the 95% range), and the latter is also consistent with noise (as most of the ALL/ANT trends are within the 95% range of noise trend). Can I then say the observed signal is also consistent with noise? I am pointing this out because there is a need to clearly define what detection means and what attribution means, and why you say ALL/ANT signals are different from noises. Without these, it is impossible to interpret your results (and your interpretation as well). Also, what do you mean by good or poor separation between a signal and noise (e.g., line 444)? There needs to be some definition to characterize.

Thank the reviewer for raising this concern. The center of the Gaussian distributions of the ALL and ANT signals were always close to or greater than the upper bound of the 95% CI of the noise (Fig. 3). This means almost or more than 50% of the ALL and ANT signals were outside the 95% CI of the noise. Therefore, most of the ALL and ANT signal trends do not fall within the 95 range of the noise trend, and the logic problem does not exist. We appreciate the reviewer for raising awareness to the fact that we did not write out any objective criteria for consistency/inconsistency. We thus added the explicit criteria to the Methods section lines 636-645 (tracked version lines 723-732):

“When a detected pseudo-observation signal was within the 95% CI of an external forcing’s signals on the same fingerprint, we considered the detected signal attributable to the external forcing, except when the external forcing did not result in signals that were distinguishable from the noise. Identifying the exception case involved comparing the 95% CIs of the external forcing’s signals and the noise. The situations in this study usually fell into two cases: (1) nearly or more than half of the 95% CI of the signals of the external forcing fell outside the 95% CI of the noise, (2) the two 95% CIs overlapped almost completely, with very similar upper and lower bounds. In the second case, we deemed the external forcing to not have resulted in distinguishable signals.”

We also noted these criteria when describing the results in lines 195-198 and 213-216 (tracked version 208-212 and 233-237):

“Also, in these detected months, nearly or more than half of the ALL, ANT, GHGAER, and GHG signals were above the upper bound of the 95% CIs of the noise (**Error! Reference source not found.h-k**), indicating that these forcings resulted in distinguishable signals from the noise.”

“However, the upper and lower bounds of the 95% CIs of the AER signals were generally close to the upper and lower bounds of the 95% CIs of the noise (**Error! Reference source not found.m-p, u-x**), indicating that the effects of AER on SSI changes were too small to warrant attribution.”

3) The method section introduced signal estimate for the model responses that involves $Z_{f_2f_1}$ and authors indicated f_2 and f_1 can be the same or different. But when you described AER is not detected, did you project both observations and AER to fingerprint of ALL (in this case f_2 is different from f_1), or both observations and AER to AER fingerprint, or observation to ALL fingerprint and AER to AER fingerprint?

Thank the reviewer for raising this question. We did two analyses: in the first, we projected both observations and AER to the ALL fingerprint; and in the second, we projected both observations and AER to the AER fingerprint. To clarify the relationship between the subscripts f_1 , f_2 and the adjectives (ALL/GHG/AER ...), we now clarified the terminology in the SI lines 171-172 and 182-184 (tracked version lines 200-201 and 212-216):

“Then, the linear least squares trend $G_{f_1o}(t_0, m, L)$ over the time series $Z_{f_1o}(m, t)$ was the pseudo-observation signal on the f_1 fingerprint.”

“Then ... we calculated the linear least squares trends $G_{f_1o}(t_0, m, L)$ for the time series $Z_{f_2f_1di}(m, t)$, and called the trend $G_{f_2f_1di}(t_0, m, L)$ the f_2 signal on the f_1 fingerprint.”

We also edited all the text and captions to ensure it is always clear on which fingerprint the signals or S/N ratios were projected unless they were generic signals or S/N ratios.

4) I appreciate the authors' explanation regarding how they did the regression to obtain the constrained S/N ratio. I now understand what they did. But the description of this important detail cannot be found anywhere in the paper or in the supplement.

The description of the emergent constraint method was placed in section 2.3, lines 224-244 of the previous revised manuscript. We see this was brief and probably difficult to find for a reader. Therefore, we added substantially more details, and placed the new description in the Methods and SI to hopefully make them clearer to the readers. These are in the revised manuscript lines 656-701 (tracked version 748-807) and SI section 4. They reflect the materials already presented in the previous response letter and the previous revised manuscript. We do not reproduce the description here because it is very long.